# Characteristics of sub-10 nm particle emissions from in-use commercial aircraft observed at Narita International Airport

Nobuyuki Takegawa[1], Yoshiko Murashima[2], Akihiro Fushimi[3], Kentaro Misawa[1], Yuji Fujitani[3], Katsumi Saitoh[3,4], and Hiromu Sakurai[2]

[1]Department of Chemistry, Graduate School of Science, Tokyo Metropolitan University, Hachioji, Tokyo 192-0397, Japan.
[2]National Institute of Advanced Industrial Science and Technology, Tsukuba, Ibaraki 305-8563, Japan.
[3]National Institute for Environmental Studies, Tsukuba, Ibaraki 305-8506, Japan.
[4]Environmental Science Analysis and Research Laboratory, Hachimantai, Iwate 028-7302, Japan.

*Correspondence to*: Nobuyuki Takegawa (takegawa@tmu.ac.jp)

**Abstract.** The characterization of ultrafine particle emissions from jet aircraft equipped with turbofan engines, which are commonly used in civil aviation, is an important issue in the assessment of the impacts of aviation on climate and human health. We conducted field observations of aerosols and carbon dioxide ($CO_2$) near a runway at Narita International Airport, Japan, in February 2018. We used an ultrafine condensation particle counter (UCPC) and a condensation particle counter (CPC) with unheated and 350°C heated operation modes to investigate the contributions of sub-10 nm size ranges to the total and the non-volatile particle number concentrations. The performance of the 350°C heated mode was tested in the laboratory to verify the consistency with existing methods for non-volatile particle measurements. We also used a scanning mobility particle sizer with unheated and 350°C heated modes and an engine exhaust particle sizer for the measurements of particle number size distributions. Spiked increases in the particle number concentrations and $CO_2$ mixing ratios were observed to be associated with the directions of wind from the runway, which can be attributed to diluted aircraft exhaust plumes. We estimated the particle number emission indices (EIs) for discrete take-off plumes using the UCPC, CPC, and $CO_2$ data. The median values of the total and the non-volatile particle number EIs for diameters larger than 2.5 nm as derived from the UCPC data were found to be $1.1 \times 10^{17}$ and $5.7 \times 10^{15}$ kg-fuel$^{-1}$, respectively. More than half the particle number EIs were in the size range smaller than 10 nm for both the total and the non-volatile particles in most of the cases analyzed in this study. The significance of sub-10 nm size ranges for the total particles in the diluted plumes was qualitatively consistent with previous studies, but that for the non-volatile particles was unexpected. Possible factors affecting the similarities and differences compared with the previous findings are discussed.

## 1 Introduction

Civil aviation has grown rapidly as a result of global economic development. Consequently, the impacts of aircraft emissions on climate and human health have been recognized as an important issue (ICAO, 2017; Masiol and Harrison, 2014; Stacey, 2019; and references therein). The characterization of ultrafine particles (UFPs; diameters of < 100 nm) is key

to understanding the environmental impacts of aircraft emissions because the particle number and mass emissions from jet engines are often dominated by the UFP size range (Masiol and Harrison, 2014; Stacey, 2019; and references therein).

The primary importance of aviation-produced aerosol particles in assessing the climate impacts is the formation of contrail cirrus clouds from soot or black carbon (BC) emitted at aircraft cruising altitudes (Kärcher and Voigt, 2017; Kärcher, 2018). Furthermore, aircraft emissions can significantly affect the number concentrations of Aitken mode particles in the upper troposphere (Wang et al., 2000; Lee et al., 2010; Righi et al., 2013, 2016). Righi et al. (2013, 2016) showed that the impacts of aviation on aerosols were sensitive to the parameterization of nucleation-mode particles (<~20 nm) in their simulation model. In our view, the contribution of aircraft emissions to the number concentration of nucleation or Aitken mode particles relative to that of other sources (new particle formation and surface emission sources) is poorly understood.

The health impacts of UFPs, although they are not specific to aircraft emissions, have been extensively studied by many researchers (Oberdörster et al., 2005; Ohlwein et al., 2019, and references therein). UFPs can be efficiently deposited in the nasal, tracheobronchial, and alveolar regions in the human respiratory system, and the uptake and translocation (physical clearance) of solid UFPs such as soot into the blood and lymph circulation could be an important pathway (Oberdörster et al., 2005). Sub-10 nm particles could be efficiently deposited in the olfactory mucosa, and the subsequent translocation of solid particles along the axons of the olfactory nerve might be a concern (Oberdörster et al., 2005). A recent study suggested that the surface reactivity of aviation-induced soot particles may increase with decreasing particle size (Jonsdottir et al., 2019). However, the health impacts of UFPs emitted from aircraft have not been well established, as pointed out by Ohlwein et al. (2019).

A number of experiments have been performed at engine-test facilities and under real-world conditions to investigate gaseous and particulate emissions from aircraft equipped with turbofan engines, which are commonly used in civil aviation (e.g., Hagen et al., 1998; Petzold et al., 1999, 2005; Kärcher et al., 2000; Brock et al., 2000; Herndon et al., 2008; Westerdahl et al., 2008; Onasch et al., 2009; Kinsey, 2009; Timko et al., 2010, 2013; Lobo et al., 2015a, 2015b; Moore et al., 2017a, 2017b; Kinsey et al., 2019; Yu et al., 2017, 2019; Durdina et al., 2019). The key findings from the previous studies include the following two points. First, significant formation and evolution of volatile particles with diameters smaller than 10 nm can take place during plume expansion, depending on the sulfur content of the fuel, the age of the plume, and the ambient conditions (Petzold et al., 1999, 2005; Kärcher et al., 2000; Brock et al., 2000, Onasch et al., 2009; Timko et al., 2010, 2013). Second, non-volatile particles, which are assumed to be equivalent to soot or BC particles, primarily have sizes in the range larger than ~10 nm (geometric mean diameter (GMD) of ~20–60 nm in particle number size distributions) under various operating conditions, with the largest GMDs under ~100% engine-thrust conditions (Petzold et al., 1999; Lobo et al., 2015a, 2015b; Moore et al., 2017a; Durdina et al., 2019). Zhang et al. (2019) recently proposed a global-scale aviation emission inventory for BC particles by integrating the existing datasets of non-volatile particle emission indices (EIs).

Alongside these scientific studies, the International Civil Aviation Organization (ICAO) has authorized a new regulatory standard for the mass and number emissions of particles emitted from aircraft engines (ICAO, 2017). In the method for measuring non-volatile particle number concentrations described in the Aerospace Recommended Practice

(ARP) 6320, issued by the Society of Automotive Engineers (SAE-ARP6320) (SAE, 2018), the number concentrations of aerosol particles with diameters larger than 10 nm are measured downstream of a volatile particle remover (VPR) heated to 350°C. The SAE-ARP6320 has been developed for the certification of jet engine emissions and may not be directly compared with ambient measurement data. Some of the above-mentioned studies employed sampling methods equivalent to SAE-ARP6320 for field measurements of particulate emissions from in-use aircraft (Lobo et al., 2015a).

Direct measurements of UFPs behind jet engines (either in engine test cells or aircraft hangers) can provide systematic emission data as a function of the engine thrust and fuel types from selected jet engines under well-controlled conditions. A common issue in measuring UFPs behind jet engines in the previous scientific studies and in the regulatory standard is the significant loss of particles in long sampling tubes and/or the VPR when fresh aircraft exhaust plumes are sampled (Kinsey, 2009; Lobo et al., 2015a, 2015b; Durdina et al., 2019). Although corrections for particle loss have been extensively

evaluated and carefully considered for quantifying particle number concentrations, the absolute values of the correction factors and relative errors associated with the corrections tend to be larger for smaller particles (Kinsey, 2009; Lobo et al., 2015a, 2015b; Durdina et al., 2019). Furthermore, the large uncertainty in measuring the particle number size distributions for diameters smaller than 20 nm by using mobility size spectrometers is also of great concern (Wiedensohler et al., 2012). These technical issues should be properly considered for better characterization of aircraft exhaust particles.

Field measurements of advected (diluted) aircraft exhaust plumes near runways are not optimal for obtaining systematic emission data, whereas potential artifacts associated with long sampling lines and/or high concentrations of condensable materials can be reduced by this approach. Furthermore, a variety of exhaust plumes from different types of in-use aircraft engines can be collectively characterized by field measurements near runways. Considering that the accessibility to platforms for sampling fresh engine exhausts (engine test cells, aircraft hangers, and runways) is generally restricted, these

approaches should be complementarily selected for better characterizing UFP emissions from aircraft. Consistent integration of the data is also important for constructing reliable emission inventories from the aviation sectors for the global troposphere (cruising altitudes), where the accessibility to sampling platforms is extremely limited.

We conducted field measurements of UFPs near a runway at Narita International Airport (NRT), Japan. We used multiple instruments for the measurements of particle number concentrations and size distributions and carefully investigated

the performance and consistency of these instruments. The purpose of the present study was to investigate the emission characteristics of sub-10 nm particles from commercial aircraft operating under real-world conditions.

## 2 Experimental

### 2.1 Field observations

The field measurements were performed using two containers placed at an observation point ~180 m from the centerline

of runway A (~140 m from the edge of the runway) at NRT between February 5 and 26, 2018 (Fushimi et al., 2019). Fig. 1 shows an approximate layout of NRT with the location of the observation point. The aerosol instruments used for the field

measurements consisted of an ultrafine condensation particle counter (UCPC; Model 3776, TSI, Inc., Shoreview, MN, USA; $d_{50}$ = 2.5 nm), a condensation particle counter (CPC; Model 3771, TSI; $d_{50}$ = 10 nm), a scanning mobility particle sizer (SMPS; Model 3080, TSI) equipped with a differential mobility analyzer (long DMA; Model 3081, TSI) and a CPC (Model 3022A; TSI; $d_{50}$ = 7 nm), an engine exhaust particle sizer (EEPS; Model 3090, TSI), and two sets of cascade impactor samplers (Nano MOUDI II; Model 125B, MSP Corp., Shoreview, MN, USA). The UCPC and CPC used for the field measurements were the same as those used for airborne measurements in our previous studies (Takegawa and Sakurai, 2011; Takegawa et al., 2014, 2017, 2020a), except for the dilution/heater sections described below. The specifications of the UCPC and CPC were similar to those of the TSI CPCs used for aircraft emission measurements near active runways in earlier studies, which included CPC 3007 ($d_{50}$ = 10 nm; Westerdahl et al., 2008), CPC 3022A ($d_{50}$ = 7 nm; Westerdahl et al., 2008; Herndon et al., 2008; Moore et al., 2017a), and CPC 3775 ($d_{50}$ = 4 nm; Moore et al., 2017a). It should be noted that the overall detection efficiencies for particles with diameters of ~10 nm may significantly depend on the configurations of the sampling tubes rather than a small difference in the $d_{50}$ values. The penetration efficiencies of particles through the sampling tubes in our system are described in Section 3.1 and the Supplement. The combination of CPCs of different detectable size ranges has been used for aircraft emission measurements at cruising altitudes (Brock et al., 2000; Moore et al., 2017b), whereas the combination of a CPC and nano-SMPS/EEPS has been used for measurements near runways (e.g., Westerdahl et al., 2008; Herndon et al., 2008; Moore et al., 2017a). We used the former method to characterize sub-10 nm particles considering its high temporal resolution and relatively large uncertainties in mobility size spectrometers for sub-20 nm size range (Wiedensohler et al., 2012). Number concentrations of aerosol particles with diameters larger than 2.5 nm or 10 nm, as measured by the UCPC or CPC, are referred to as $N_{2.5}$ or $N_{10}$, respectively. Details of the performance of the UCPC and CPC are described in Section 3.1. The other instruments included a carbon dioxide ($CO_2$) monitor (Model LI-840, Li-Cor Biosciences, Lincoln, NE, USA), a nitrogen oxides ($NO_x$) monitor (APNA-370, Horiba, Japan), meteorological sensors, and a video camera for monitoring aircraft passages. We used the data from the UCPC, CPC, SMPS, EEPS, and $CO_2$ monitor recorded during February 15–22, 2018 for the present analysis. Although $NO_x$ data are useful for characterizing aircraft emissions, we did not use those data because the response time was not sufficient for capturing rapid changes in the concentration in aircraft plumes.

Fig. 2a illustrates a schematic diagram of the sampling setup for the UCPC, CPC, SMPS, and $CO_2$ monitor. Ambient air was drawn into the container through a stainless-steel tube (ID: 10.7 mm; length: ~3 m) and was split into a bypass flow connecting to a piston pump and sample flows for aerosol and $CO_2$ measurements. The total flow rate through the main tube (sum of the bypass and sample flows) was ~20 L min$^{-1}$. The aerosol sample flow was diluted by a factor of ~5 by using particle-free air (2 L min$^{-1}$) to extend the concentration range measured by the UCPC and CPC, which would otherwise have been limited by particle-coincidence effects. The diluted sample flow was then passed through a stainless-steel evaporation tube (ID: 7.5 mm; tube length: 320 mm; heated section: 200 mm) for heated sampling or through a bypass tube for unheated (room temperature) sampling. A thermocouple sensor was attached to the upstream part of the evaporation tube for temperature control. The three-way valve was switched between the unheated and heated modes every 8 h. We set the heater

temperature to 250°C during February 7–9, 150°C during February 11–13 and 22–23, and 350°C during February 15–21. We used the data obtained during the unheated and 350°C heated modes for the present analysis. The characterization of the 150 and 250°C heated modes is ongoing and will be presented elsewhere. A copper tube (ID: 7.5 mm; length: ~600 mm) between the evaporation tube and the valve was used to cool the heated sample air. The sampling method for 350°C heated

$N_{10}$ approximately corresponded to the method for measuring non-volatile particles described in SAE-ARP6320 (SAE, 2018), although we did not use a thermal denuder or a catalytic stripper so as to reduce the particle diffusion loss. The estimation of potential artifacts is described in Section 2.2. The tube downstream of the three-way valve was split into individual sample flows for the UCPC (1.4 L min⁻¹), CPC (1 L min⁻¹), and SMPS (0.3 L min⁻¹). Electrically conductive tubes (ID: 4.8 mm; Part 3001788, TSI) were used for the connections between the splitter and the instruments. The SMPS was disconnected

from the dilution/heater section after February 18, and the flow rate settings were changed accordingly. Note that the flow rates for calculating the dilution factor were calibrated by using the standard flowmeter at the National Institute of Advanced Industrial Science and Technology (AIST). The dilution factor was estimated to be ~3.8 before February 18 and ~6.1 after that date. The residence time in the evaporation tube (320 mm, 350°C) was ~0.15 s before February 18 and ~0.17 s after that. These values are shorter than the requirement for the residence time (>0.25 s) described in SAE-ARP6320 (SAE, 2018),

although the actual temperature profile of the sample air inside the evaporation tube and its downstream was not measured. The performance of the evaporation tube was tested using tetracontane ($C_{40}H_{82}$) particles, as described in Section 2.3. After the SMPS had been disconnected, the CPC 3022 was connected directly to the main sampling tube upstream of the dilution/heater section. This permitted an overall validation of the dilution method, because the maximum concentration range of the CPC 3022 was extended to $10^7$ cm⁻³ by photometric detection. Although we used the dilution method to reduce

the effects of particle coincidence, we frequently observed high particle number concentrations (>$10^5$ cm⁻³) downstream of the dilution section. The effects of particle coincidence were corrected using the methods described in our previous studies (Takegawa and Sakurai, 2011; Takegawa et al., 2017). The correction factors were as high as ~40% at nominal (uncorrected) particle number concentrations of $5 \times 10^5$ cm⁻³ and $1.2 \times 10^5$ cm⁻³ for the UCPC and CPC, respectively. The uncertainty in the corrections would become larger at even higher concentrations.

As mentioned earlier, the measurements of particles below 20 nm using mobility size spectrometers might include large uncertainties (Wiedensohler et al., 2012). The major sources of uncertainties in the SMPS measurements originated from the corrections for the charging efficiency and Brownian diffusion, the latter generally being more significant for nanoparticles. The Aerosol Instrument Manager (AIM) software provides correction tools for these factors. We have found that the number size distributions at diameters below 20 nm show non-negligible differences for different versions of the software. We used

AIM version 9.0 for the present analysis, with the diffusion loss correction enabled. The effects of the AIM diffusion correction on the derived number size distributions are described in Section S2 of the Supplement.

   The EEPS was operated independently from the UCPC/CPC/SMPS inlet system (Fig. 2b) and it measured the unheated particle number size distributions during the entire period. The sample flow rate of the EEPS was 10 L min⁻¹. A copper tube (ID: 10 mm, length: ~2 m), electrically conductive tubes (ID: 4.8 and 7.9 mm; total length: ~1 m; Part 3001788, TSI), and a

glass manifold (inner diameter: 40 mm; total length: 600 mm) were used for the ambient sampling. A bypass pump (flow rate: 10 L min$^{-1}$) was used to reduce the particle diffusion loss. We used the default instrument matrix for the EEPS, which may underestimate size and concentration of particles larger than ~75 nm (Wang et al., 2016). Although the particle diameter range detectable by the EEPS extended from ~6 to 520 nm, our laboratory experiments showed that the EEPS may significantly underestimate particle number concentrations below 10 nm. Reduced detection efficiencies of sub-10 nm particles can also be inferred from the EEPS data obtained by other researchers (Moore et al., 2017a). The evaluation of the EEPS is ongoing and will be presented elsewhere. In this study we used the EEPS data for particle diameters larger than 10 nm.

The $CO_2$ instrument was calibrated by using two $CO_2$ standards (397.2 and 1032 parts per million by volume (ppmv)) twice a day during the measurement period. The injection of the $CO_2$ standards was performed automatically by using solenoid valves. We found that the sensitivity of the instrument was generally stable during the measurement period.

The $N_{2.5}$, $N_{10}$, EEPS, and $CO_2$ data were obtained every 1 s, and the SMPS data for particle diameters from 15 to 660 nm were obtained every 5 min (scanning time: 3 min). When we observed spiked increases in the particle number concentrations and $CO_2$ mixing ratios in aircraft plumes, the timing of the detection of the concentration peaks did not exactly match between the individual instruments. Because this was likely caused by differences in the response time of the instruments and the delay time in the sampling tubes, the data was shifted accordingly (<10s).

The overall penetration efficiencies of particles through the sampling tubes were estimated by using the theoretical formulae proposed by Gormley and Kennedy (1949). Details of the calculation procedures are given in Section S1 of the Supplement. The penetration efficiency for the UCPC and CPC sampling line (unheated mode; from the top of the inlet to the flow splitter) was estimated to be 70%, 87% and 94% for particle diameters of 5, 10, and 20 nm, respectively. The penetration efficiency through the evaporation tube and the detection efficiencies of the UCPC and CPC are important parameters for the data interpretation and are discussed in detail in Section 3.1. The penetration efficiency for the SMPS sampling line (unheated mode; from the top of the inlet to the SMPS) was estimated to be 79% and 85% for 15 and 20 nm, respectively, and that for the EEPS sampling line (from the top of the inlet to the EEPS) was estimated to be 94% and 98% for 10 and 20 nm, respectively.

The particle diffusion loss during sampling is an important issue for the quantification of UFPs, as mentioned in Section 1. The corrections for the penetration efficiencies through the sampling tubes and the detection efficiencies (see Section 3.1 for details) were not incorporated in the UCPC and CPC data presented in Sections 3.2.1–3.2.4 because the actual size distributions in the sub-10 nm size range were uncertain. Furthermore, the corrections for the penetration efficiencies were not incorporated in the SMPS and EEPS data presented in Sections 3.2.2–3.2.3 for consistency with the UCPC/CPC data: i.e., they include only the default (internal) corrections for the individual instruments. We considered the effects of particle diffusion loss as systematic uncertainties. We also considered the penetration efficiencies through the sampling tubes and the detection efficiencies for estimating the size distributions of particle number EIs in Sections 3.2.5 and 4.

## 2.2 Potential artifacts

Potential artifacts due to the nucleation of gaseous compounds vaporized from particles in the evaporation tube (hereafter referred to as nucleation artifacts) were evaluated. Predicting the nucleation rates requires an estimate of the supersaturation of nucleating compounds, which is highly uncertain. Here we evaluate the growth rate of nucleated clusters under the given condition. The upper limit of this effect can be estimated by assuming that the number concentration of the vaporized compounds remains constant after a certain period of time (~1 s) and that all the compounds are condensed into nucleated clusters (e.g., sulfuric acid). In real situations, only a small fraction of the vaporized compounds might contribute to the condensational growth of particles because of possible increases in their saturation vapor pressures by thermal decompositions and their deposition onto the inner surface of the sampling tube.

Let us assume a condensable material (number concentration of molecules: $c$; molecular weight: $MW$) in the gas-phase vaporized from particles. The original mass concentration in the particle phase before heating, $m$, is expressed as follows:

$$m = \frac{MW}{N_A} c \qquad (1)$$

where $N_A$ is the Avogadro number. Assuming a kinetic regime, the growth of nucleated particles is governed by the following expression (Seinfeld and Pandis, 2006):

$$\frac{\pi}{2} \rho D_p^2 \frac{dD_p}{dt} = \frac{MW}{N_A} \frac{\pi}{4} D_p^2 \bar{v} \alpha (c - c_{eq}) \qquad (2)$$

where $\rho$ is the particle density, $D_p$ is the diameter of nucleated particles, $\bar{v}$ is the mean thermal velocity of condensable gas molecules, $\alpha$ is the accommodation coefficient, and $c_{eq}$ is the equilibrium number concentration. Assuming $\alpha = 1$ and $c_{eq} = 0$ (maximum molecular flux), we obtain the expression:

$$\frac{dD_p}{dt} = \frac{\bar{v}}{2\rho} m \qquad (3)$$

If we consider sulfuric acid as a condensable material, $dD_p/dt$ (m s$^{-1}$) can be approximated as $0.1m$ at 350°C (maximum gas temperature), where $m$ is expressed in units of kg m$^{-3}$. The residence time for particle growth after the evaporation tube is estimated to be <1 s. The residence time of 1 s and the dilution factor of 5 lead to the estimation that the maximum particle growth is 1 nm at an ambient mass concentration of 50 μg m$^{-3}$. The minimum particle diameter at which the detection efficiency becomes zero is ~2 and ~7 nm for the UCPC and CPC, respectively (Takegawa and Sakurai, 2011; Takegawa et al., 2017). If we assume an initial cluster size of 1 nm (approximately the critical cluster size of sulfuric acid), the effects of

the artifacts on $N_{2.5}$ would be significant in aircraft exhaust plumes with relatively high concentrations (>50 µg m$^{-3}$), and those on $N_{10}$ (and SMPS) would be significant at very high concentrations (>300 µg m$^{-3}$). This estimation does not change significantly if we assume high-molecular-weight organic compounds from jet engine lubrication oil (slower thermal velocities and smaller particle densities compared with sulfuric acid).

Other potential artifacts may originate from the condensational growth of non-volatile particles (or residual particles downstream of the evaporation tube) smaller than the detectable size range of the UCPC (diameter < 2 nm). For example, residual particles with diameters of ~2 nm may grow to ~3 nm at an ambient mass concentration of 50 µg m$^{-3}$. However, this effect is significant only in the presence of a large fraction of non-volatile particles with diameters below 2 nm.

### 2.3 Laboratory experiments

The accuracy of the measurements of particle number concentrations was the key issue in this study and thus was evaluated in the laboratory at AIST before and after the field measurements. We mainly used the data obtained after the field measurements because they were more comprehensive than those obtained before the measurements. The test items included the size-resolved detection efficiencies of the UCPC and CPC, the penetration efficiency of non-volatile particles through the dilution/heater section, and the removal efficiency of volatile compounds through the evaporation tube.

An electrospray aerosol generator (EAG; Model 3480, TSI), a combustion aerosol standard (CAST; Matter Engineering, AG, Wohlen, Switzerland) with a tube furnace for thermal treatment at 350°C, and a custom-made tube furnace for supplying condensable vapors were used to generate polydisperse aerosol particles for the calibrations. We used sucrose particles supplied from the EAG (particle diameter range: 6–15 nm) for the detection efficiency experiment, non-volatile propane soot particles supplied from the CAST (15–100 nm) for the penetration efficiency experiment, and tetracontane particles supplied from the tube furnace particle generator (15–30 nm) for the removal efficiency experiment. The tube furnace downstream of the CAST was used to remove organic compounds internally or externally mixed with soot. We used a volatility tandem DMA with a heater temperature of 350°C to confirm that the thermal treatment efficiently removed organic compounds; i.e., there was no significant difference in the mobility diameter of soot particles between the two DMAs.

The experimental apparatus for measuring the size-resolved detection efficiencies of the UCPC and CPC was similar to that used in our previous studies (Takegawa and Sakurai, 2011; Takegawa et al., 2017), except that the sampling line was longer for the CPC. The length of the sampling line for the CPC was ~90 cm, which was limited by the instrument layout in the observation rack. We set the particle diameters at 15, 30, 50, and 100 nm for testing the penetration efficiencies and at 15 and 30 nm for testing the removal efficiencies, in accordance with the SAE-ARP6320 protocol. The flow rate through the evaporation tube was set at 2.4 L min$^{-1}$. We used an aerosol electrometer (AE; Model 3068B, TSI) and a reference CPC (ref CPC; Model 3775, TSI) as reference instruments. These were calibrated with the standard AE maintained by AIST for particle number concentrations at ambient pressures in the size range of 10–300 nm.

We also tested the removal efficiencies of tetracontane particles for diameters of 30 and 50 nm in the laboratory at Tokyo Metropolitan University (TMU). We used another CPC (Model 3772; TSI), which is essentially the same as CPC Model 3771, for the detection of residual particles downstream of the evaporation tube because the conditions for the CPC 3771 were not optimized during the experiments at TMU. The flow rate through the evaporation tube was set at either 2.4 or 265 2.7 L min$^{-1}$ to test the dependency of the removal efficiencies on the flow rate. Considering that the polydisperse size distributions of the tetracontane particles generated at TMU were rather broad, we also measured the removal efficiencies of doubly charged particles (43 nm for 30 nm and 72 nm for 50 nm).

## 3 Results

### 3.1 Laboratory experiments

270    Fig. 3a shows the size-resolved detection efficiencies of the UCPC and CPC measured at AIST. The detection efficiencies for the UCPC and CPC were empirically estimated by using our previous calibration results for the CPC (Takegawa and Sakurai, 2011), the manufacturer specifications for the UCPC (Takegawa et al., 2017), and the penetration efficiencies in the instrument (for the UCPC; Wimmer et al. (2013)) and the sampling lines. The penetration efficiencies were calculated by using the theoretical formulae proposed by Gormley and Kennedy (1949). Further details of the 275 empirically estimated detection efficiencies are given in Fig. S1 in the Supplement. We found a good agreement between the experimental data and the estimated detection efficiencies for both the UCPC and CPC.

Fig. 3b shows the penetration and detection efficiencies of non-volatile soot particles through the 350°C heated mode sampling tubes as measured by the UCPC. The temperature was set at room temperature (~19–21°C) and at 350°C for comparison. The penetration and detection efficiencies include the penetration efficiency through the dilution/heater section 280 and the detection efficiency of the UCPC shown in Fig. 3a. Based on the specifications in the SAE-ARP6320 protocol, the penetration efficiency through a VPR should be equal to or higher than 30, 55, 65, or 70% for particle diameters of 15, 30, 50, and 100 nm, respectively. Our data showed that the penetration efficiency was well above the requirements for all diameters tested. It was rather difficult to perform a simple theoretical prediction of the penetration efficiency at 350°C because the actual temperature profile of the sample air was uncertain. Therefore, the penetration efficiency curve at 350°C 285 (not shown in Fig. 3) was determined by scaling the calculated curve at 20°C with the experimental values at 350°C and at room temperature for larger diameters (>30, 50, and 100 nm).

The SAE-ARP6320 protocol specifies that the removal efficiencies of tetracontane particles in a VPR should be higher than 99.9% for particle diameters of 15 and 30 nm. For the removal efficiency test at AIST, we confirmed that the removal efficiencies of tetracontane particles for diameters of 15 and 30 nm were >99.9% (remaining fraction of <0.1%) for both the 290 UCPC and CPC. For the removal efficiency test at TMU, we first confirmed that there was no significant difference exceeding the experimental uncertainties between the two flow rate settings (2.4 and 2.7 L min$^{-1}$) for both the UCPC and CPC, and we mainly used the results for 2.7 L min$^{-1}$ (shorter residence time). Table 1 presents the experimental results. The

remaining fractions for 30 nm particles were found to be 0.3 (+0.2/–0.0) % and <0.1% for the UCPC and CPC, respectively. The difference between the results from the AIST and TMU experiments for the UCPC was not well identified, and we took the results from the TMU experiments for conservative estimates. The remaining fractions for 50 nm particles were found to be 5 (+4/–1) % and <0.1% for the UCPC and CPC, respectively. The remaining fraction for 73 nm was not quantified because of low particle number concentrations (thus the multiple-charge correction for 50 nm was minor). These results suggested that about 5% of the 50 nm tetracontane particles might not have fully vaporized but shrunk to sizes between 2.5 and 10 nm downstream of the evaporation tube. Potential influences on the interpretation of the ambient data are discussed later

## 3.2 Field observations

### 3.2.1 Particle number concentrations

Fig. 4 shows time series of $N_{2.5}$ and $CO_2$ obtained on February 15, 2018. The inlet valve was switched from the unheated to the 350°C heated mode at 16:00 local time (LT). The wind directions were east or east-southeast for Fig. 4a (average ± standard deviation (1σ) of the wind speeds were $3.9 ± 0.7$ m s$^{-1}$) and east or east-northeast for Fig. 4b (average ± 1σ of the wind speeds were $4.3 ± 0.6$ m s$^{-1}$), indicating that relatively stable winds from the direction of the runway were dominant. The ambient temperature was ~10°C for Fig. 4a and ~8°C for Fig. 4b. We did not observe a significant enhancement of aerosols or $CO_2$ when the air parcels originated from the opposite direction from the runway during aircraft operating times (06:00-23:00 LT). The spikes in the concentrations of aerosol particles and in $CO_2$ mixing ratios shown in Fig. 4 can be interpreted as resulting from diluted exhaust plumes from individual aircraft movements. The observed values of $N_{2.5}$ (and $N_{10}$) for the 350°C heated mode (Fig. 4b) were significantly lower than those for the unheated mode (Fig. 4a) when comparing plumes with similar enhancement levels of $CO_2$. The depletions of $N_{2.5}$ (and $N_{10}$) for the 350°C heated mode were much larger than those expected from the difference in the penetration efficiencies of aerosol particles between the unheated and the 350°C heated modes (see Fig. 3). Although the characteristics of the plumes shown in Figs. 4a and 4b cannot be directly compared, the similarity in the wind conditions suggests that the ages of the plumes were comparable (~50 s). These results imply that most of the aerosol particles in the observed plumes were volatile. We have previously shown that jet engine lubrication oil was the major source of aerosol particles with diameters ranging from ~10 to 30 nm in air parcels observed during the measurement period (Fushimi et al., 2019). A key point in Fig. 4 is that the number fraction of particles with diameters ranging from 2.5 to 10 nm (red area) was generally larger than that of particles with sizes above 10 nm (gray area), for both the unheated and 350°C heated modes. The significance of sub-10 nm size ranges for the total particles in diluted plumes ($10^1$–$10^2$ m from the jet engines) was qualitatively consistent with the findings from previous studies conducted on the ground (e.g., Petzold et al., 2005; Lobo et al., 2012, 2015b) or in-flight at cruise altitudes (Petzold et al., 1999; Kärcher et al., 2000; Brock et al., 2000), but that for non-volatile particles found in this study was unexpected.

Fig. 5a is a scatterplot of 1 s averaged values of $N_{10}$ versus those of $N_{2.5}$ for the 350°C heated mode for the time period from 17:00 to 23:00 on February 15, 2018, the same data as used in Fig. 4b. Data obtained on February 21, 2018 at slightly different flow rate settings (see Section 2.1) are also plotted to show the reproducibility of the results. The 1:1 correspondence line ($N_{10} = N_{2.5}$) represents the state in which all particles are included in the size range larger than 10 nm, and deviations below the 1:1 line indicate an increase in the fractions of sub-10 nm particles. These results show that many data points lie below the 1:2 line (sub-10 nm fraction of >50%) when $N_{2.5}$ exceeds $10^5$ cm$^{-3}$. This point is more quantitatively found in the median values for each concentration bin (Fig. 5b). Some data points in Fig. 5a were found on the 1:10 correspondence line (sub-10 nm fraction of 90%). The deviations from the 1:1 line would have been even larger had we considered the size-dependent loss of particles in the sampling line shown in Fig. 3.

### 3.2.2 Size distributions of total particles

Fig. 6 shows time series of total particle number concentrations measured by the CPC and EEPS and the particle number and volume size distributions ($dN/d\log D$ and $dV/d\log D$) measured simultaneously by the EEPS and unheated SMPS for selected time periods on February 15, 2018. The total particle number concentrations measured by the CPC ($d_{50} = 10$ nm) and EEPS (10–520 nm) showed reasonable agreement, which confirms the overall consistency between these two independent measurements. The time periods were carefully selected for investigating the effects of rapid changes in the particle number concentration during each SMPS scan on the derived particle size distribution. The SMPS needed ~30 s to scan the major population of particle number concentrations in the aircraft plumes (<30 nm). Considering possible delays in the detection timing due to the residence time in the SMPS sampling tubes (<5 s), the EEPS data were averaged over 40 s around the timing of the SMPS scan corresponding to the diameter range of 15–30 nm (i.e., averaged from 10 s before to 30 s after the onset of each SMPS scan). For the SMPS scan at 14:10 LT, the particle number concentration varied by a factor of ~2 during the first 30 s scan time, but there was no systematic increase or decrease in the particle number concentration. We found a reasonable agreement between the EEPS and SMPS (within a factor of ~2). For the SMPS scan at 14:20 LT, the particle number concentration decreased significantly during the 30 s scan time and then started to increase rapidly afterwards. The resultant number size distribution was likely affected by these concentration changes. These results demonstrate that the SMPS data can be used to investigate the characteristics of particle size distributions within specific size ranges by selecting the appropriate time windows. Note that the SMPS data tend to exhibit higher concentrations (by a factor of ~2) at smaller diameters (<20 nm) compared with the EEPS data in both cases. The difference cannot be explained by the uncertainty due to the penetration efficiencies through the sampling tube (as indicated by error bars in Fig. 6b). The difference may be due to uncertainties in the default (internal) correction algorithms in the SMPS (see Fig. S2 in the Supplement). Nevertheless, the overall size dependency (i.e., increasing particle number concentrations with decreasing particle diameters below 20 nm) is consistent between the SMPS and EEPS data.

In the enhancement events at 14:10 and 14:20 LT in Fig. 6, the EEPS data indicate that the particle number size distribution functions below 50 nm showed significant increases compared with that in the non-enhancement event (14:00

LT). The particle volume size distribution function below 50 nm exhibited moderate increases in the enhancement events, but the total integrated particle volume concentrations were largely affected by accumulation-mode particles (>50 nm) in the background air. A similar feature was also found for other time periods (Fushimi et al., 2019; Misawa et al., manuscript in preparation).

### 3.2.3 Size distributions of non-volatile particles

Fig. 7 shows the particle number and volume size distributions measured simultaneously by the EEPS and the 350°C heated SMPS for selected time periods with and without enhancements of aerosol particle number concentrations on February 15, 2018. The time periods were selected so that there would be no systematic increase or decrease in the particle number concentrations during the first 30 s scan time of the SMPS. Similarly to Fig. 6, the particle number size distribution functions below ~50 nm in the enhancement events showed significant increases compared to those in the non-enhancement event for both the EEPS and the 350°C heated SMPS. The particle number size distribution functions below ~50 nm measured by the 350°C heated SMPS were smaller by more than an order of magnitude compared to those measured by the EEPS, indicating that the aircraft exhaust particles were mostly volatile below 50 nm. Our laboratory experiments suggest that there might remain residues of >50-nm particles in the sub-10 nm size range after the evaporation tube (the remaining fraction was ~5% for 50 nm tetracontane particles). However, they were likely negligible compared with the observed sub-10 nm non-volatile particles because the particle number concentrations for a diameter range of >50 nm measured by the EEPS (unheated) were far below the observed values of the 350°C heated $N_{2.5} - N_{10}$ (Fig. 7a). In addition, residues of 30–50 nm particles in the sub-10 nm size range after the evaporation tube were likely minor compared with the observed sub-10 nm non-volatile particles considering the sharp decrease in the $dN/d\log D$ values from 30 to 50 nm as measured by the EEPS (the remaining fractions were ~0.3% and 2% for 30 and 43 nm tetracontane particles, respectively).

The effects of nucleation artifacts (Section 2.2) might be a major concern but were likely small under the observation conditions because the mass concentrations of aerosol particles in the aircraft plumes inferred from Fig. 7 were much lower than the threshold concentration (50 μg m$^{-3}$) described in Section 2.2. In fact, we did not find any systematic increases in the $N_{2.5}/N_{10}$ ratios for the 350°C heated mode with increasing total particle volume concentrations derived from the EEPS (not shown). Furthermore, the particle number and volume size distributions derived from the 350°C heated SMPS (Figs. 7b and c) exhibited no indication of the presence of an additional mode resulting from nucleation artifacts (Section 2.2). The 350°C heated $dN/d\log D$ functions in Fig. 7b exhibited gradual increasing trends with decreasing particle diameters from 100 to 15 nm. This feature is unlikely to be explained by the artificial growth of particles downstream of the evaporation tube because the mass concentrations of aerosols in the plumes would not have been sufficient to yield particle growth exceeding 1 nm (Section 2.2).

### 3.2.4 Particle emission indices for take-off plumes

The temporal variations and number size distributions of aerosol particles clearly indicate that the observed air parcels were significantly affected by aircraft emissions under appropriate wind conditions. However, aircraft emissions from various cycles of take-off, landing, and idling may have been mixed in the atmosphere. We calculated the enhancements of $N_{2.5}$, $N_{10}$, and $CO_2$ above the background levels (referred to as $\Delta N_{2.5}$, $\Delta N_{10}$, and $\Delta CO_2$, respectively), and extracted discrete plumes during the take-off phases by setting some criteria for these parameters (for details, see Section S3 and Figs. S3–S5 in the Supplement). By using the data obtained on February 15, 16, 20, 21, and 22, we identified 132 discrete plumes for the unheated mode and 63 for the 350°C heated mode. Particle number and volume size distributions ($dN/d\log D$ and $dV/d\log D$) for the take-off plumes were derived from the EEPS data (unheated conditions), and the enhancements of $dN/d\log D$ and $dV/d\log D$ above the background levels were calculated using the same method as for $\Delta N_{2.5}$.

The $\Delta N_{2.5}/\Delta CO_2$ and $\Delta N_{10}/\Delta CO_2$ ratios were converted into particle number EIs by assuming a $CO_2$ EI value of 3160 g of $CO_2$ per kilogram of fuel (Moore et al., 2017a), as shown in Table 2. The corrections for the penetration and detection efficiencies of the UCPC and CPC were not incorporated into the above estimates because the actual size distributions were uncertain; these values should therefore be regarded as lower limits. This point is further discussed in Section 4.1. The arrival time of the plumes was estimated to be ~30–120 s by considering the wind directions and speeds, corresponding to the transport distance of ~180–370 m from the engine exits to the observation point (Section S3 of the Supplement). We did not find a systematic dependence of the $\Delta N_{10}/\Delta CO_2$ and $\Delta N_{2.5}/\Delta CO_2$ ratios on the estimated arrival time of the plumes (Fig. S5 in the Supplement), suggesting that the variation in plume ages did not yield significant biases in our results. We did not perform a detailed classification of the plumes by jet engine types because information on the engines of the aircraft observed at NRT was not available. We performed a simple classification of the plumes by aircraft model (Table S2 in the Supplement), but did not find significant differences between the different types of aircraft.

The median total and non-volatile EI($N_{2.5}$) values, which likely cover the major size range of the aircraft emissions, were found to be $1.1 \times 10^{17}$ and $5.7 \times 10^{15}$ kg-fuel$^{-1}$, respectively. The difference in these values (a factor of ~20) is interpreted as the average contribution of volatile particles. We defined the sub-10 nm fraction as $1 - \Delta N_{10}/\Delta N_{2.5}$ for the identified discrete plumes. The median and the central 50 percentile range of the sub-10 nm fraction for the unheated mode were found to be 0.63 and 0.53–0.70, respectively, and those for the 350°C heated mode were 0.54 and 0.44–0.72, respectively (Table 2). The significance of sub-10 nm particles for the total and the non-volatile particles, which was shown in Sections 3.2.1–3.2.3 (Figs. 4–7) as a case study, was also found in the statistical analysis of the take-off plumes. Considering that the penetration efficiencies of particles through the sampling tubes (Fig. S1 in the Supplement) and the evaporation tube (Fig. 3) tended to decrease with decreasing particle diameters below 10 nm, these results suggest that more than half the total and the non-volatile particle number EIs in the aircraft take-off plumes were found in the size range smaller than 10 nm in most cases.

Previous studies have shown that the particle number EIs can vary significantly depending on the engine type, the
engine thrust, the fuel sulfur content, the plume age, and the ambient conditions (Petzold et al., 1999, 2005; Kärcher et al.,
2000; Brock et al., 2000; Onasch et al., 2009; Timko et al., 2010). The sampling setup (deployed instruments and sampling
location relative to the runways) and the analysis procedures (discrete plume analysis) of this study are similar to those of
Lobo et al. (2012, 2015b) and Moore et al. (2017a). Therefore, it is worthwhile discussing the similarities and differences
between those studies and our results. The fuel sulfur content is an important parameter for the comparison with other studies.
We do not have information on the sulfur content of the fuel that was actually used at NRT. Instead, we analyzed fuel
samples (Jet A-1) provided by a jet fuel company in Tokyo (Ishinokoyu, Co. Ltd.) (Saitoh et al., 2019b). We obtained a total
of five samples between August 2017 and August 2018. The sulfur content of the fuel samples ranged from 30.4 to 440 parts
per million by weight (ppmw). We assume that these values are representative of the sulfur content of jet fuels commercially
available in Tokyo during the observation period.

Lobo et al. (2012) reported particle number and mass EIs measured 100–350 m downwind of the runways at Oakland
International Airport in August 2005. They used a fast particulate spectrometer (DMS500, Cambustion) to measure the
particle number size distributions for diameters ranging from 5 to 1000 nm. The total particle number EIs for various types
of engines under take-off conditions ranged from $4 \times 10^{15}$ to $2 \times 10^{17}$ kg-fuel$^{-1}$, which inclusively covered the 25–75
percentile range of the total EI($N_{2.5}$) from our measurements. The fuel sulfur content was estimated to be 240–395 ppmw.

Lobo et al. (2015b) reported the particle number and mass EIs measured near the jet engine exits and 100–350 m
downwind of the runways at Hartsfield-Jackson Atlanta International Airport in September 2004. The former corresponds to
the non-volatile particle EIs and the latter corresponds to the total particle EIs. They used a DMS500 to measure the particle
number size distributions. The non-volatile particle number EIs for a JT8D-219 engine were $\sim 1 \times 10^{16}$ kg-fuel$^{-1}$ under 85%
and 100% thrust conditions, which were close to the 75 percentile value of the non-volatile EI($N_{2.5}$) from our measurements.
The total particle number EIs for various types of engines under take-off conditions ranged from $7 \times 10^{15}$ to $9 \times 10^{17}$ kg-fuel$^{-1}$, which again inclusively covered the 25–75 percentile range of the total EI($N_{2.5}$) from our measurements. The information
on the fuel sulfur content was not provided.

Moore et al. (2017a) reported the particle number and volume EIs for take-off plumes measured 400 m downwind of
the runway at Los Angeles International Airport (LAX) in May 2014. They used a CPC 3775 ($d_{50}$ = 4 nm) for the
measurements of total particles, a CPC 3022A ($d_{50}$ = 7 nm) for non-volatile particles, and an EEPS for the measurements of
size distributions. We calculated the median values (and the 25–75 percentile range) of the total and the non-volatile particle
number EIs provided by Moore et al. (2017a)  to be 4.6 (3.1–5.8) $\times 10^{16}$ and 2.1 (1.1–3.6) $\times 10^{15}$ kg-fuel$^{-1}$, respectively. The
median and 25–75 percentile range of the total and the non-volatile EI($N_{10}$) from our measurements showed good agreement
with those from Moore et al. (2017a). Although Moore et al. (2017a) used different types of CPCs, the detection efficiency
curves for the CPC 3771 and 3022A were likely similar according to the manufacturer specifications. Furthermore, the EEPS
data from Moore et al. (2017a) showed that the contributions from sub-10 nm particles to the total particle number EIs were
relatively small (see Section 3.2.5 for details). Therefore, the good agreement between their results and the EI($N_{10}$) values

from our measurements would be reasonable. The sulfur content of the jet fuel samples collected at LAX ranged from 620 to 1,780 ppmw (average value of 1,180 ppm).

Timko (2010) showed that the total particle number EIs in moderately diluted plumes (measured by a CPC 3022A), which were dominated by volatile particles, exhibited a relatively weak dependence on the fuel sulfur content (<1,500 ppmw) under high thrust conditions for various types of engines. Provided that the fuel sulfur content was likely below ~1,500 ppmw for the previous studies (Lobo et al., 2012, 2015b; Moore et al., 2017a) and this study, it might not be the major factor affecting the variability in the emissions of volatile particles among these studies (at least for diameters larger

than ~10 nm).

### 3.2.5 Size distributions of total particles for take-off plumes

The enhancements of $dN/d\log D$ and $dV/d\log D$ above the background levels derived from the EEPS data were converted to the corresponding EI values ($d$EI($N$)/$d\log D$ and $d$EI($V$)/$d\log D$). As described in Section 2.1, the corrections for the penetration efficiencies through the sampling tubes (Fig. S1) were considered for the data conversion. Fig. 8 shows the

derived $d$EI($N$)/$d\log D$ and $d$EI($V$)/$d\log D$ values for the take-off plumes. The size distributions of the particle number and volume EIs for the take-off plumes observed at LAX (Moore et al., 2017a), which are characterized by bimodal log-normal distributions (nucleation and soot modes), are shown for comparison. Other previous studies also reported that aircraft emissions can be characterized by distinct bimodal size distributions under high engine thrust conditions (e.g., Kinsey, 2009; Lobo et al., 2012, 2015b; Yu et al., 2017, 2019).

The total and non-volatile particle number EIs derived from the UCPC and CPC fell in the same range as those from the previous studies for take-off plumes under real-world operating conditions (Lobo et al., 2012, 2015b; Moore et al., 2017a), as described in Section 3.2.4. However, the characteristics of the size distributions appeared to be significantly different. The mode diameters of the $d$EI($N$)/$d\log D$ for the nucleation mode reported by the previous studies were 10–20 nm in most cases. We compared the $d$EI($N$)/$d\log D$ values between each size bin (midpoint diameters of 10.8, 12.4, 14.3, … nm) for the

individual take-off plumes shown in Fig. 8. We found that the $d$EI($N$)/$d\log D$ at 10.8 nm exhibited the highest value for ~98% of the plumes. We also found that the $d$EI($N$)/$d\log D$ at 10.8 nm was more than two times larger than that at 14.3 nm for 79% of the plumes. These results suggest that the $d$EI($N$)/$d\log D$ values tended to increase with decreasing particle diameters around 10–20 nm and that the mode diameters of the $d$EI($N$)/$d\log D$ for the nucleation mode were smaller than 10 nm for the majority of the plumes observed in this study. Our results also suggest that the peak values of the $d$EI($N$)/$d\log D$ for the

nucleation mode were much larger than those reported by Moore et al. (2017a), in which the contributions from the sub-10 nm size range to the total particle number EI size distributions were relatively small.

The uncertainties in the $d$EI($N$)/$d\log D$ and $d$EI($V$)/$d\log D$ values at larger particle diameters (>100 nm) were considerably large in our data because they were significantly affected by the accumulation-mode particles in the background air, and also because the default instrument matrix of the EEPS might underestimate size and concentrations of

particles larger than ~75 nm (Wang et al., 2016). Nevertheless, it is likely that the non-volatile particle number and volume

EIs originating from soot-mode particles (>20 nm) were much smaller than those reported by the previous studies (Lobo et al., 2012, 2015b; Moore et al., 2017a). A correction for the EEPS size bin by a scaling factor of 1.1, as was done by Moore et al. (2017a), would increase the particle volume concentrations by a factor of 1.33, which does not fill the gap between the previous studies and our results.

## 4 Discussion

### 4.1 Estimates of particle number EI size distributions for take-off plumes

We estimated the possible particle number size distributions for the total and the non-volatile particles constrained by the UCPC and CPC observations and investigated the consistency with the EEPS and SMPS measurements. We assumed log-normal number size distributions with various geometric mean diameters (GMDs) and geometric standard deviations (GSDs). We calculated the number fraction of sub-10 nm particles $(1 - \Delta N_{10}/\Delta N_{2.5})$ by integrating the number size distributions weighted by the penetration and detection efficiency curves:

$$\text{Calculated sub-10 nm fraction} = 1 - \left( \int \eta_{\text{CPC}} \left( \frac{dN}{d\log D} \right) d\log D \right) \Big/ \left( \int \eta_{\text{UCPC}} \left( \frac{dN}{d\log D} \right) d\log D \right) \qquad (4)$$

$$\text{Calculated EI}(N_{2.5}) = \int \eta_{\text{UCPC}} \left( \frac{d\text{EI}(N)}{d\log D} \right) d\log D \qquad (5)$$

where $\eta_{\text{CPC}}$ and $\eta_{\text{UCPC}}$ are the overall penetration and detection efficiencies for the CPC and UCPC, respectively. We assumed monomodal size distributions for $dN/d\log D$ and $d\text{EI}(N)/d\log D$ considering the shapes of the observed size distributions for the total and the non-volatile particles (Figs. 6 and 7). The overall penetration efficiency, which included the sampling tubes from the rooftop and the dilution/heater sections (Figs. 2 and 3b), was calculated based on the theoretical formulae by Gormley and Kennedy (1949). The GMD and GSD values assumed in Eq. (4) were also used to calculate the $d\text{EI}(N)/d\log D$ in Eq. (5). The calculated EI($N_{2.5}$) was compared with the median EI($N_{2.5}$) derived from the observations to retrieve the absolute values of $d\text{EI}(N)/d\log D$ and EI($N$).

Fig. 9 shows the calculation results for the total particles. GMD values of <10 nm are needed to explain the median sub-10 nm fraction derived from the plume analysis (0.63) with a realistic range of GSDs (1.2–1.6). For example, the GMD value of 7.9 nm is obtained if we assume a GSD value of 1.6 (or the GMD value of 8.6 nm is obtained if we assume a GSD value of 1.4). The retrieved $d\text{EI}(N)/d\log D$ values were found to be consistent with those derived from the EEPS (Fig. 8), which supports the validity of our estimate. The retrieved EI($N$) value was $1.7 \times 10^{17}$ and $1.6 \times 10^{17}$ kg-fuel$^{-1}$ for (GMD, GSD) = (7.9 nm, 1.6) and (8.6 nm, 1.4), respectively. These values were larger by a factor of ~1.5 compared to the median

total EI($N_{2.5}$) value, suggesting that the particle number EIs derived from the unheated UCPC might underestimate the true particle number EIs by a factor of ~1.5.

Fig. 10 shows the calculation results for the non-volatile particles. Similarly to the total particles, GMD values of <10 nm would also be needed to explain the median sub-10 nm fraction derived from the plume analysis (0.54). The GMD value of 9.0 nm is obtained if we assume a GSD value of 1.6 (or the GMD value of 9.5 nm is obtained if we assume a GSD value of 1.4). The retrieved $d$EI($N$)/$d$log$D$ values were found to be consistent with those derived from the SMPS, although the size distribution data to support the validity of this estimate are limited. The retrieved EI($N$) value was $1.0 \times 10^{16}$ and $0.95 \times 10^{16}$ kg-fuel$^{-1}$ for (GMD, GSD) = (9.0 nm, 1.6) and (9.5 nm, 1.4), respectively. These values were larger by a factor of ~1.7 compared to the median non-volatile EI($N_{2.5}$) value, suggesting that the particle number EI derived from the 350°C heated UCPC might underestimate the true non-volatile particle number EIs by a factor of ~1.7.

## 4.2 Interpretation and implications

The characteristics of the total particles observed in this study are qualitatively consistent with the findings from previous studies (e.g., Petzold et al., 2005; Lobo et al., 2012, 2015b; Petzold et al., 1999; Kärcher et al., 2000; Brock et al., 2000): i.e., the total particle number EIs are dominated by volatile particles, and the sub-10 nm particles make significant contributions to the total particle number EIs in the diluted plumes ($10^1$–$10^2$ m from the jet engines). In contrast to the total particles, the significance of sub-10 nm size ranges for the non-volatile particle number EIs is unexpected. Currently, we do not have direct evidence that the observed sub-10 nm non-volatile particles are composed mainly of soot. Nevertheless, it is worthwhile considering possible mechanisms for the formation of sub-10 nm non-volatile particles. Our estimate implies the presence of very small non-volatile particles with diameters down to a few nanometers. This is not consistent with the size of the primary soot particles from jet engines estimated by using transmission electron microscopy (TEM) or laser-induced incandescence (LII) methods (e.g., Liati et al., 2014; Boies et al., 2015; Saffaripour et al., 2017, 2020), although there still remain substantial uncertainties in estimating the size of primary particles by those methods, as pointed out by Boies et al. (2015).

An alternative possibility for the significance of sub-10 nm non-volatile particles includes the potential contributions of less volatile organic matter (as compared with tetracontane) or metal compounds. Fushimi et al. (2019) showed that the mass contribution of the sum of trace elements (other than carbonaceous and sulfur compounds) was comparable to that of elemental carbon (soot) for UFP samples (~10–30 nm) collected at NRT. Saitoh et al. (2019a) reported that metal elements including Ca, Fe, Si, Mg, K, Zn, Pb, and Ni were the major compositions of these trace elements. These metal compounds might have contributed to the non-volatile particles observed in this study.

The key point in our results is that the non-volatile particle number and volume EIs originating from soot-mode particles (>20 nm) were much smaller than those reported by the previous studies for take-off plumes under real-world operating conditions (Lobo et al., 2012, 2015b; Moore et al., 2017a). This feature might be related to the significance of sub-10 nm non-volatile particles. The data presented by Moore et al. (2017a), which were obtained in 2014, exhibited lower

contributions of soot-mode particles compared to those by Lobo et al. (2012, 2015b), which were obtained in 2005 and 2004, respectively. A possible explanation for this tendency is that newer engines might emit smaller non-volatile particles as compared with older engines.

Our results have an implication for jet engine exhaust measurements by the SAE-ARP6320 method. The non-volatile $N_{10}$ data obtained from the 350°C heated CPC approximately correspond to the definition of non-volatile particles specified by SAE-ARP6320, at least in terms of the detectable size range of the particle counter and the removal efficiency of tetracontane particles (Section 3.1). If a population of non-volatile particles having a similar size distribution as shown in Fig. 10 is measured by the SAE-ARP6320 method, subtle changes in the size-dependent penetration and detection efficiencies of the measurement system and/or shifts in the mode diameter of the particle population might lead to large variabilities in the results. Further investigations are needed to quantify the size distributions of non-volatile particles for various types of engines under different conditions.

Our results also have an implication for the emission inventories of the aviation sector. Although the potential contributions of sub-10 nm particles inferred from our results likely have negligible impacts on the mass concentrations of ambient aerosol particles, they may have non-negligible impacts on the number concentrations in and around airports and also at aircraft cruising altitudes. The lifetime of sub-10 nm particles would be short near the ground level due to evaporative loss (e.g., Fushimi et al., 2008) and/or coagulation scavenging onto pre-existing larger particles, but it could be much longer under conditions of low temperatures and reduced concentrations of pre-existing aerosol particles. We propose that emissions of sub-10 nm particles from aircraft under real-world conditions should be properly considered for understanding the impacts of aviation on human health and also for developing aviation emission inventories for regional and global models.

## 5 Conclusions

We conducted field measurements of aerosols at an observation point ~180 m from the centerline of a runway at Narita International Airport. We investigated the characteristics of particle emissions from in-use commercial aircraft under real-world operating conditions, with specific focuses on the contributions of sub-10 nm size ranges to total and non-volatile (350°C heated) particles. We used the UCPC, CPC, SMPS, and EEPS for the measurements of particle number concentrations and size distributions and carefully investigated the performance and consistency of these instruments. The major conclusions are summarized below.

- The median values of the total and the non-volatile EI($N_{2.5}$), which likely cover the major size range of aircraft emissions, were found to be $1.1 \times 10^{17}$ and $5.7 \times 10^{15}$ kg-fuel$^{-1}$, respectively. The difference in these values (a factor of ~20) is interpreted as the average contribution of volatile particles. We did not find a systematic dependence of the total particle number EIs on the estimated plume age (~30–120 s). The true particle number EIs for total and non-volatile particles

might be larger by a factor of ~1.5 and ~1.7, respectively, compared to the above median values considering the penetration efficiencies through the sampling tubes and the evaporation tube.

- More than half the total and the non-volatile particle number EIs in the aircraft take-off plumes were found in the size range smaller than 10 nm for most of the cases analyzed in this study. The significance of sub-10 nm size ranges for the total particles was qualitatively consistent with previous studies, but that for the non-volatile particles was unexpected.

- The unheated UCPC, CPC, and EEPS data consistently suggest that the mode diameters of the $d\mathrm{EI}(N)/d\mathrm{log}D$ for nucleation-mode particles were smaller than 10 nm for the majority of the observed take-off plumes.

- The EEPS data suggest that the non-volatile particle number and volume EIs originating from soot-mode particles (>20 nm) were much smaller than those reported by previous studies for take-off plumes under real-world operating conditions.

The characteristics of particle emissions may significantly depend on the type of jet engine, the maintenance conditions, and the fuel sulfur content, which are not available in this study. Particle emissions may also depend on other factors including ambient pressure and temperature. These factors should be carefully considered for a more systematic comparison of different studies.

**Data availability**

The field measurement data used in this study are available at the Zenodo data repository (http://doi.org/10.5281/zenodo.4279160; data citation: Takegawa et al. (2020b)).

**Author contributions**

NT, AF, and HS designed the research; NT, AF, KM, YF, and KS performed field observations and collected data; NT, YM, and HS performed laboratory experiments; NT and YM performed data analysis; NT, AF, YF, and HS wrote the paper.

**Competing interests**

The authors declare no conflict of interest.

**Acknowledgments**

We thank Narita International Airport Corporation and Narita International Airport Promotion Foundation for their help during the field observations at Narita International Airport. We also thank Takumi Saotome at Research Institute for Environmental Strategies, Inc., Makiko Mine and Anna Nagasaki at Tokyo Metropolitan University for their help in the observations and data analysis, and Kenjiro Iida at AIST for useful advice on the evaluation of the UCPC and CPC. This

study was funded by the Environment Research and Technology Development Fund (5-1709) of the Ministry of the Environment, Japan.

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

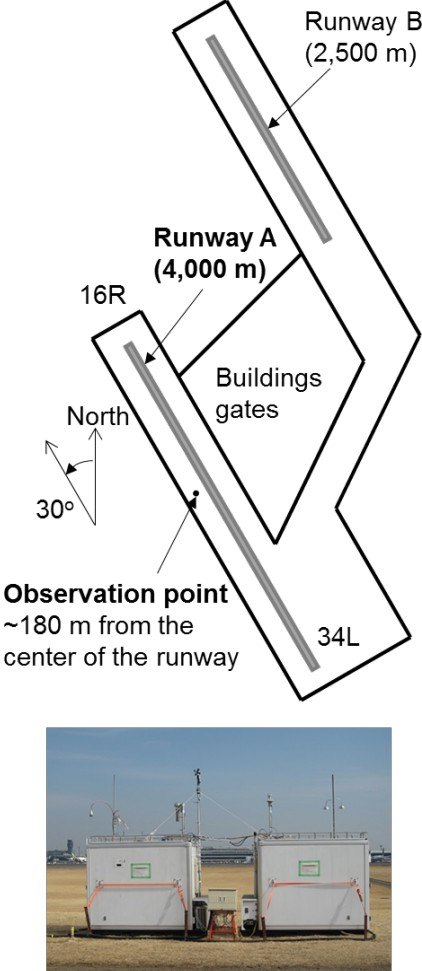

**Figure 1:** Approximate layout of Narita International Airport (NRT). The observation point was located ~180 m from the centerline of runway A (~140 m from the edge of the runway). The azimuth of runway A is 30° from the north (23° from the magnetic north). The photograph shows the containers of instruments deployed at the observation point.

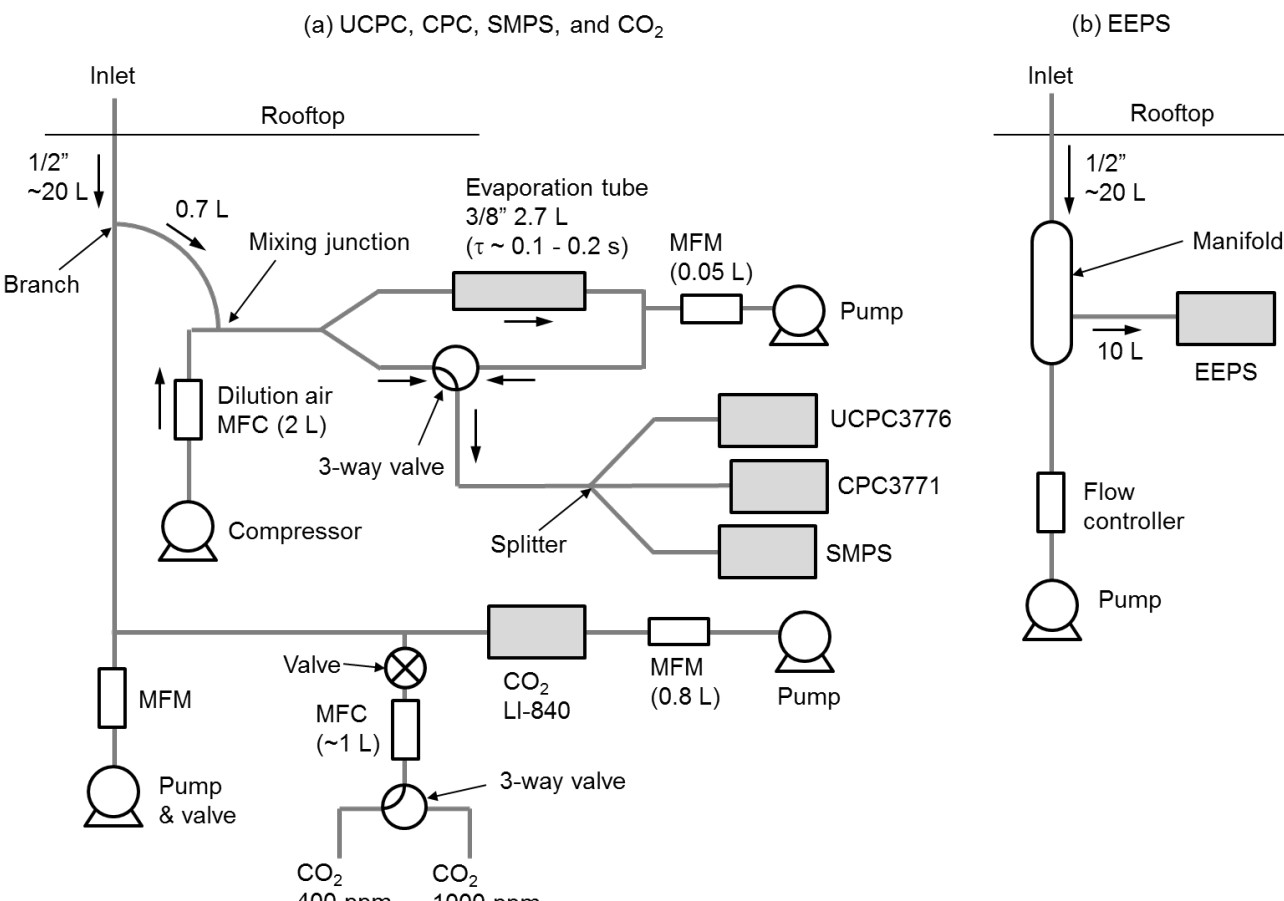

**Figure 2:** (a) Schematic of the sampling setup for the UCPC, CPC, SMPS, and $CO_2$ monitor. Ambient air was drawn into one of the two containers through a stainless steel tube and was split into a bypass flow and sample flows for aerosols and $CO_2$. The aerosol sample flow was diluted by particle-free air regulated by a mass flow controller (MFC) to extend the concentration range measured by the UCPC and CPC. The diluted sample flow was then passed through a heated stainless-steel tube (evaporation tube) for heated sampling, or a bypass tube for unheated sampling. The flow was switched between the two paths by an automated three-way valve downstream. The tube downstream of the three-way valve was split into individual sample flows for the UCPC, CPC, and SMPS. An additional small flow (~0.05 L min–1) was maintained by an orifice, a mass flow meter (MFM), and a pump to avoid the creation of a reverse stream from the evaporation tube during unheated sampling. (b) Schematic of the sampling setup for the EEPS. Ambient air was drawn into the other container through a copper tube, electrically conductive tubes, and a glass manifold. The sample flow for the EEPS was taken from the manifold. The total flow through the copper tube was maintained by a pump downstream of the manifold.

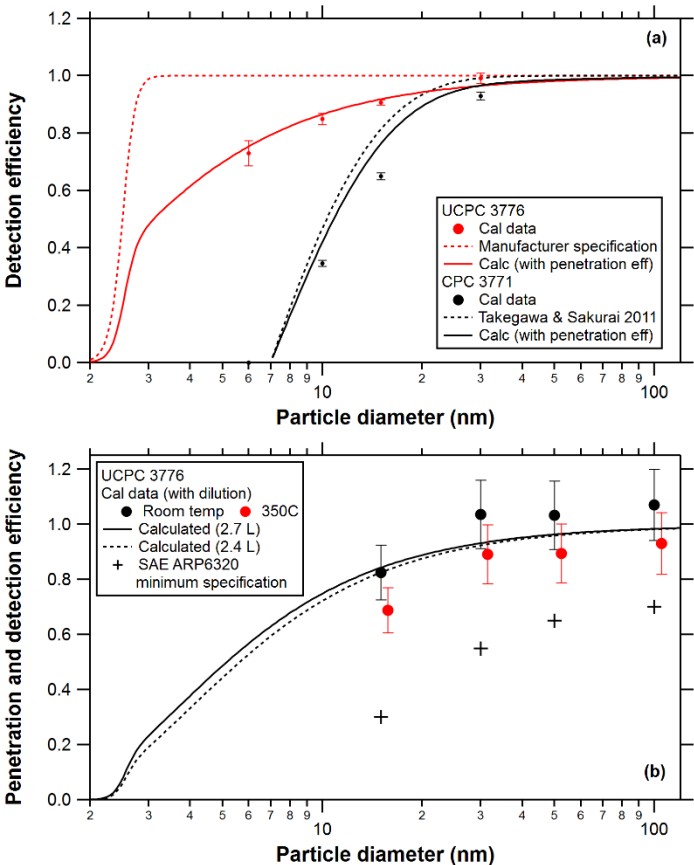

**Figure 3:** Laboratory evaluation of the performance of the UCPC and CPC. (a) Size-resolved detection efficiencies of the UCPC (red circles) and CPC (black circles) measured in the laboratory. The curves represent the empirically calculated detection and penetration efficiency curves for the UCPC and CPC: the detection efficiency of the UCPC (red dashed line); the detection efficiency of the UCPC incorporating the penetration efficiencies in the UCPC internal and sampling tube (red solid line); the detection efficiency of the CPC (black dashed line); and the detection efficiency of the CPC incorporating the penetration efficiency through the sampling tube (black solid line). See Section S1 of the Supplement for details on the definition. (b) Penetration and detection efficiency of non-volatile propane soot particles through the heater section measured at room temperature (~19–21°C; black circles) and at 350°C (red circles). The calculated curves include the penetration efficiency through the heater section and the detection efficiency of the UCPC at 20°C with the flow rate through the heater at 2.7 (solid) and 2.4 (dashed) L min$^{-1}$. The penetration efficiency curve at 350°C (not shown) was determined by scaling the calculated curve at 20°C with the experimental values at 350°C and room temperature for larger diameters (30, 50, and 100 nm). The cross symbols represent the minimum penetration efficiency of non-volatile particles specified by the SAE-ARP6320 protocol.

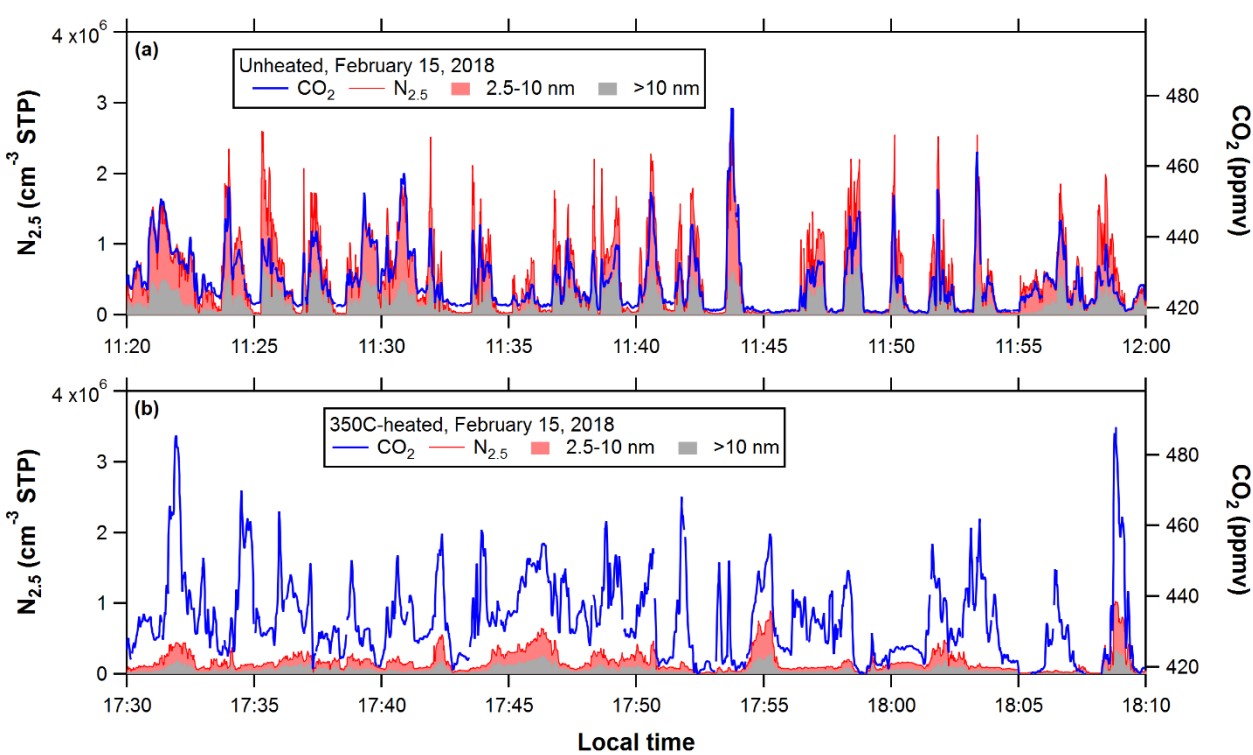

**Figure 4:** Time series of $N_{2.5}$ and $CO_2$ observed near a runway at NRT. Data for (a) unheated and (b) 350°C heated modes obtained on February 15, 2018. The red and gray areas represent the number fraction of particles with diameters ranging from 2.5 to 10 nm and that of particles with diameters above 10 nm for both the unheated and 350°C heated modes.

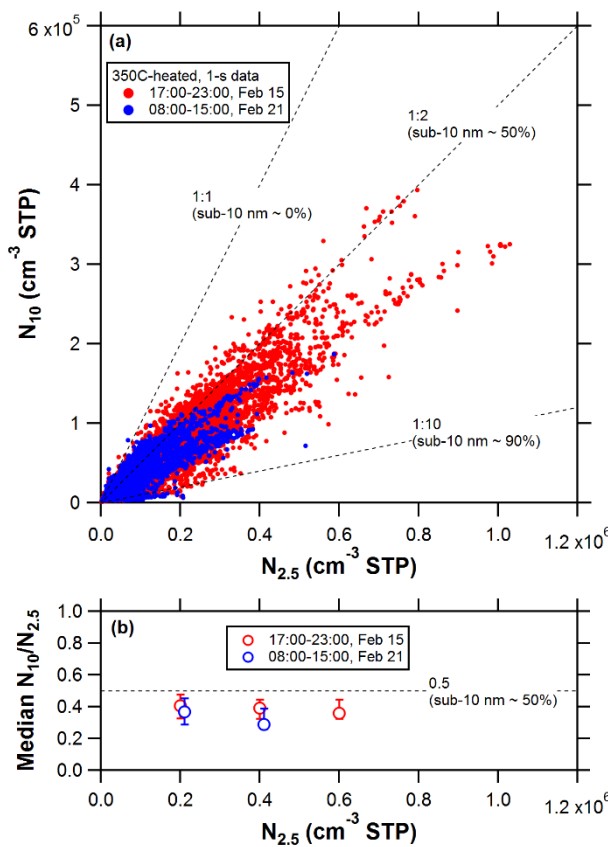

**Figure 5:** (a) Scatterplots of $N_{10}$ versus $N_{2.5}$ for the 350°C heated mode. Data obtained on February 15, 2018 (red) and February 21, 2018 (blue) are shown. The 1:1 correspondence line ($N_{10} = N_{2.5}$) represents the state in which all particles are included in the size range larger than 10 nm. (b) Median values of the $N_{10}/N_{2.5}$ ratios for the $N_{2.5}$ bin of $(1–3) \times 10^{15}$, $(3–5) \times 10^{15}$, and $(5–7) \times 10^{15}$ calculated by using the data shown in (a). The error bars represent the 25 and 75 percentile values.

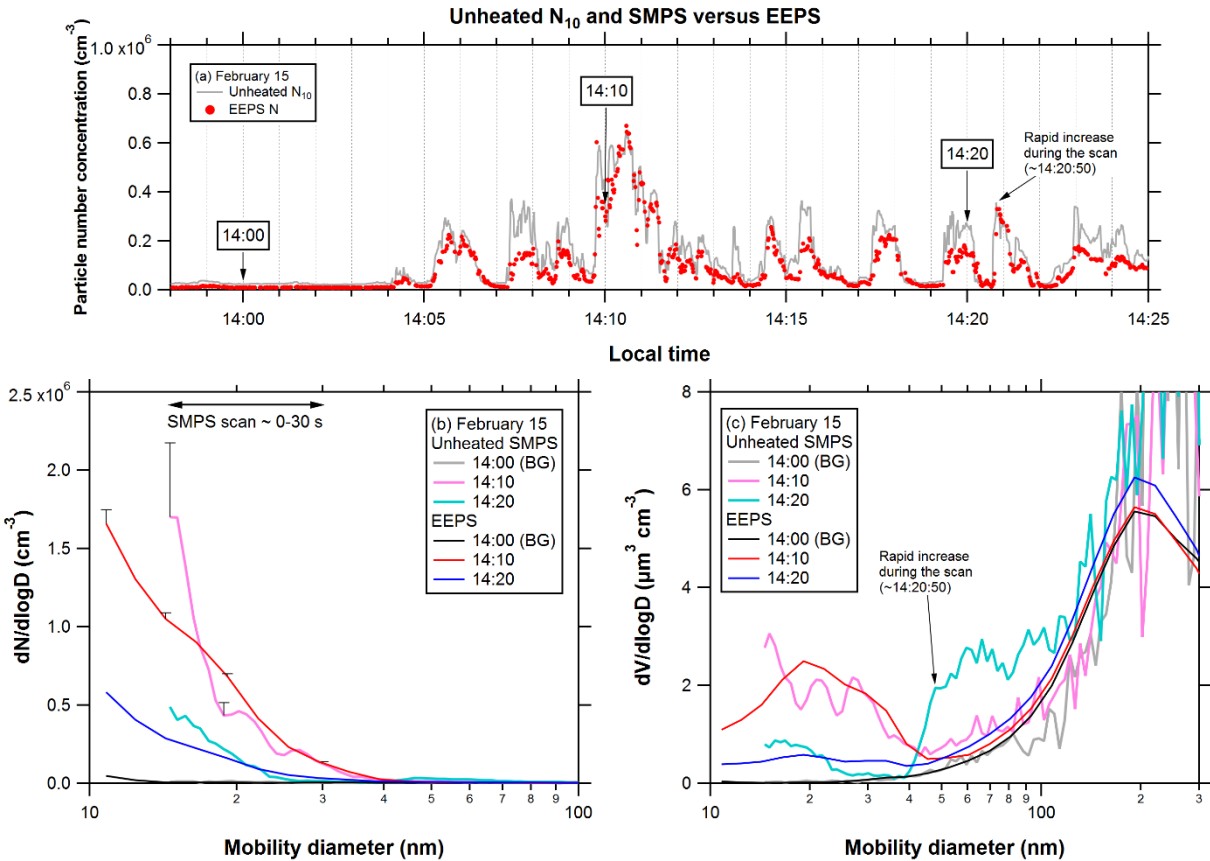

**Figure 6:** (a) Time series of unheated $N_{10}$ (gray) and total particle number concentrations derived from the EEPS data (EEPS N, red) obtained at 13:58–14:25 LT on February 15, 2018. (b) Particle number size distributions measured simultaneously by the EEPS and unheated SMPS for selected time periods indicated in (a). The EEPS data were averaged over 40 s around the timing of the SMPS scan corresponding to the diameter range of 15–30 nm (i.e., averaged from 10 s before to 30 s after the onset of each SMPS scan). "BG" denotes a time period without enhancements of the aerosols and $CO_2$. The upper ends of the error bars indicate the particle number concentrations incorporating the penetration efficiencies of particles through the sampling tubes shown in Fig. S1. (c) Same as (b) but for the particle volume size distributions.

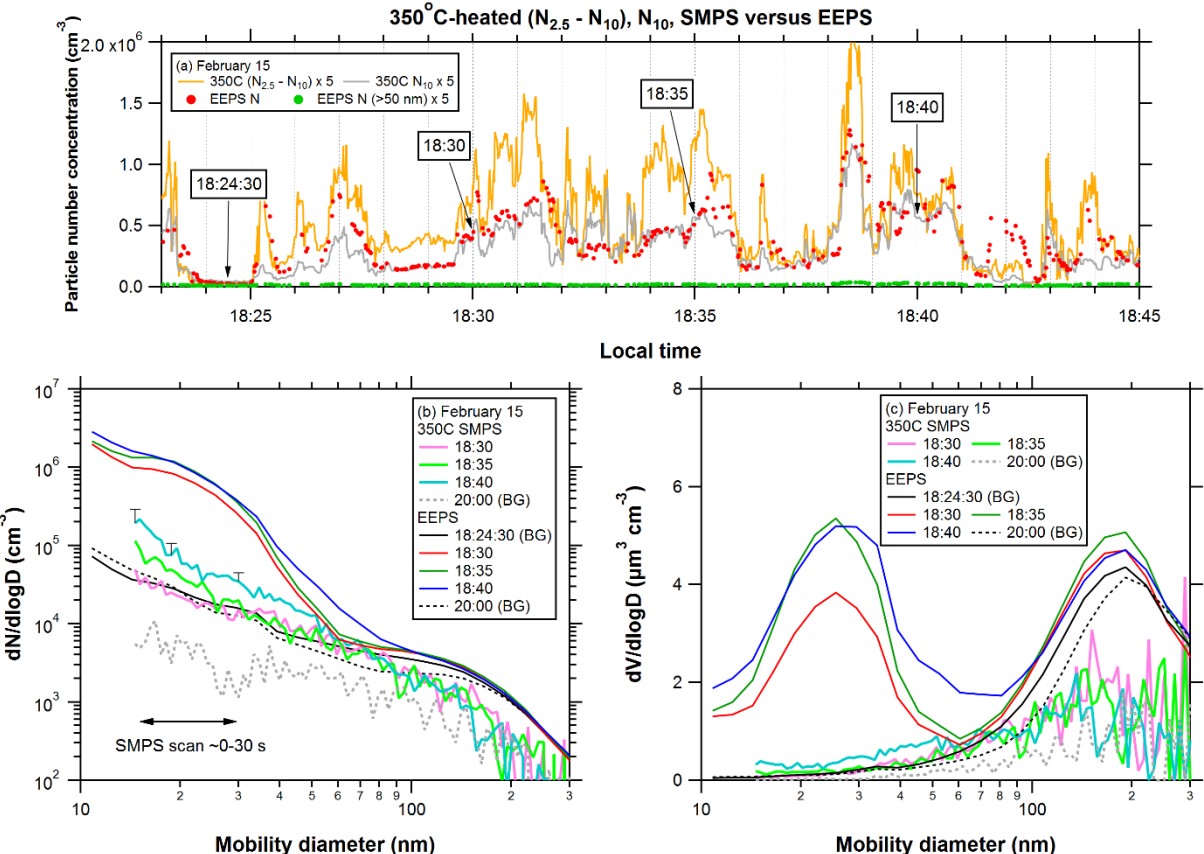

**Figure 7:** (a) Time series of 350°C heated $N_{2.5} - N_{10}$ (orange), $N_{10}$ (gray), total particle number concentrations derived from the EEPS data (EEPS N, red), and particle number concentrations for a diameter range of >50 nm measured by the EEPS (EEPS N (>50 nm), green) obtained at 18:23–18:45 LT on February 15, 2018. The 350°C heated $N_{2.5} - N_{10}$, $N_{10}$, and EEPS N (>50 nm) data are multiplied by a factor of 5 for clarity. (b) Particle number size distributions measured simultaneously by the EEPS and 350°C heated SMPS for selected time periods indicated in (a). The average method for the EEPS is the same as that used in Fig. 6b. "BG" denotes a time period without enhancements of the aerosols and $CO_2$. The upper ends of the error bars indicate the particle number concentrations incorporating the penetration efficiencies of particles through the sampling tubes and evaporation tube (Figs. 3 and S1). (c) Same as (b) but for the particle volume size distributions.

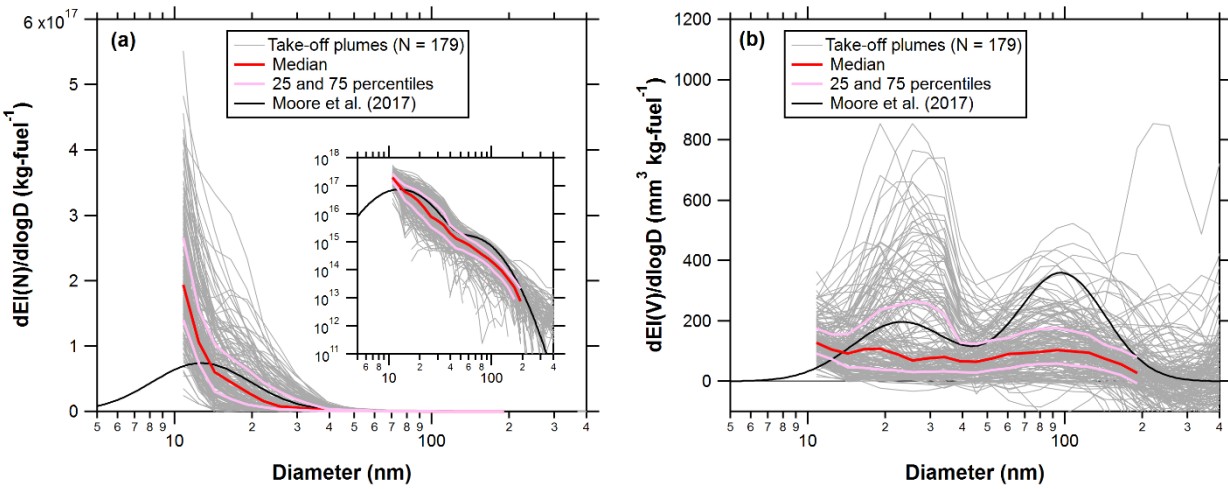

**Figure 8:** Size distributions of (a) number and (b) volume EIs derived from the EEPS data for the take-off plumes. The penetration efficiencies of particles through the sampling tubes (Fig. S1) are incorporated in these estimates. The shaded lines represent all data for the take-off plumes. The red and two pink lines indicate the median, 25, and 75 percentiles, respectively. The black line represents the average
size distributions from Moore et al. (2017a). A log-log plot of the same data is inserted in (a) for better visualization of the smaller values.

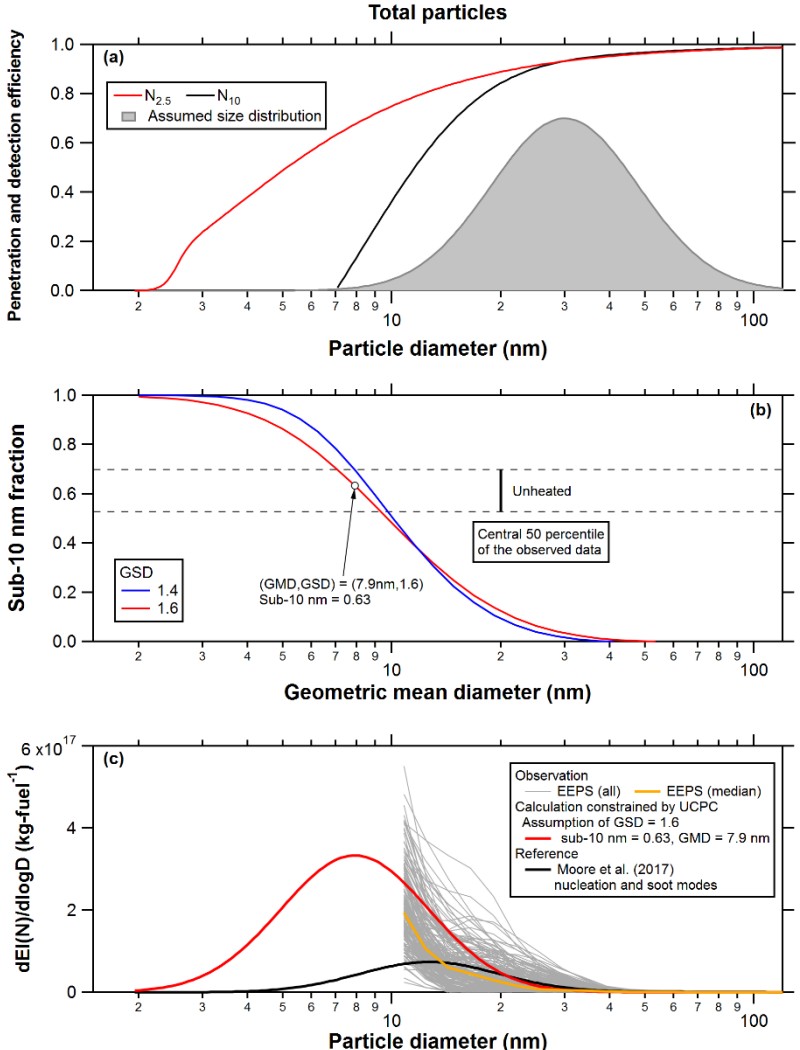

**Figure 9:** (a) Overall penetration and detection efficiencies for the unheated $N_{2.5}$ and $N_{10}$. The shaded area represents an example of the assumed particle number size distributions for calculating the number fraction of sub-10 nm particles. (b) Number fraction of sub-10 nm particles $(1 - \Delta N_{10}/\Delta N_{2.5})$ obtained by convolution of the penetration and detection efficiency curves in (a) and assumed particle number size distributions having various GMDs and GSDs. The horizontal dashed lines indicate the central 50 percentile range (between 25 and 75 percentiles) of the observed data for the unheated mode. (c) Example of the particle number EI size distributions that explains the observed medians of the $\Delta N_{10}/\Delta N_{2.5}$ ratios and EI($N_{2.5}$) for the unheated mode (red). The number EI size distributions with (GMD, GSD) = (12.7 nm, 1.56) and (61 nm, 1.48), which correspond to the average nucleation and soot mode distributions (black) reported by Moore et al. (2017a), are shown for comparison. The size distributions of particle number EIs for the take-off plumes derived from the EEPS data (Fig. 8a) are also shown.

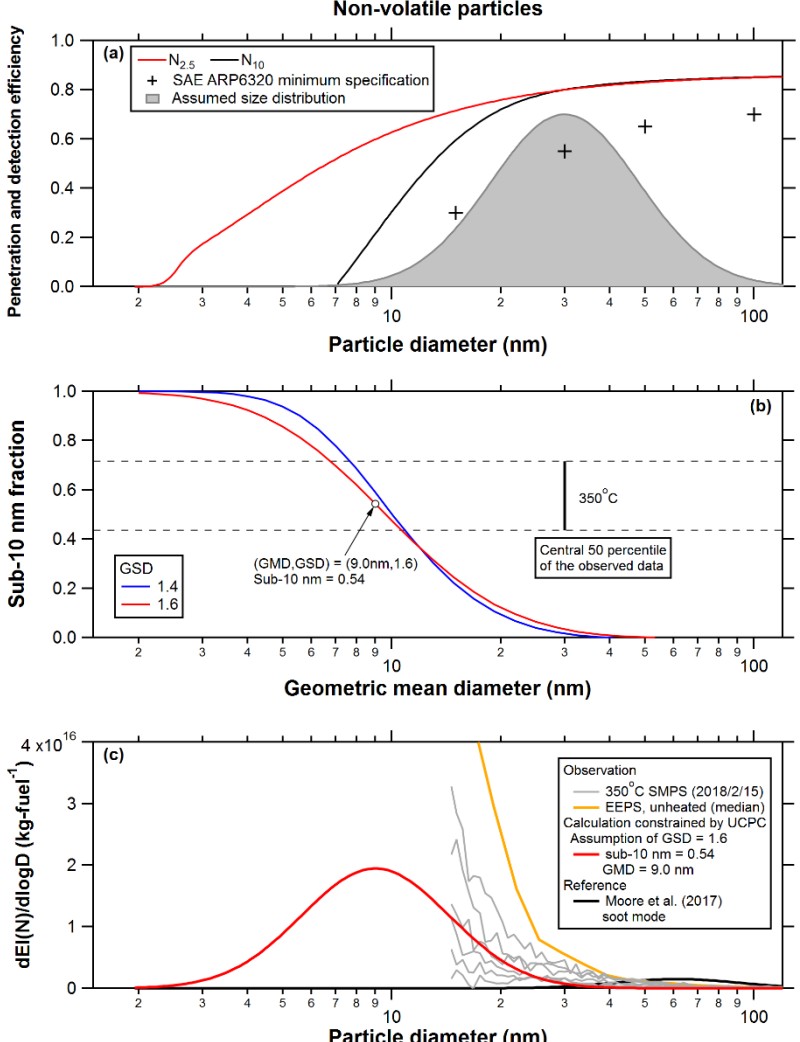

**Figure 10:** (a) Overall penetration and detection efficiencies for the 350°C heated $N_{2.5}$ and $N_{10}$. The lower limit of the penetration efficiency through the VPR specified in SAE-ARP6320 is shown for comparison. The shaded area represents an example of the assumed particle number size distributions for calculating the number fraction of sub-10 nm particles. (b) Number fraction of sub-10 nm particles (1 − $\Delta N_{10}/\Delta N_{2.5}$) obtained by convolution of the penetration and detection efficiency curves in (a) and assumed particle number size distributions having various GMDs and GSDs. The horizontal dashed lines indicate the central 50 percentile range (between 25 and 75 percentiles) of the observed data for the 350°C heated mode. The calculation results for the unheated (Fig. 9b) and 350°C heated modes were nearly identical because we assumed that the size dependency of the penetration efficiency of particles through the evaporation tube for the 350°C heated mode was the same as that for the unheated mode. (c) Examples of the particle number size distributions that can explain the observed median of $\Delta N_{10}/\Delta N_{2.5}$ ratios for the 350°C heated mode (red). The size distribution of particle number EI with (GMD, GSD) = (61 nm, 1.48), which corresponds to the average soot mode distribution (black) reported by Moore et al. (2017a), is shown for comparison. The size distributions of particle number EIs for the take-off plumes estimated from the 350°C heated SMPS on February 15, 2018 are also shown. The penetration efficiencies of particles through the sampling tubes and evaporation tube (Figs. 3 and S1) are incorporated in the SMPS estimates.

**Table 1:** Remaining fraction (%) of tetracontane ($C_{40}H_{82}$) particles measured by the UCPC and CPC downstream of the evaporation tube.

| Particle size | 30 nm | 43 nm | 50 nm |
|---|---|---|---|
| UCPC | 0.3 (+0.2/–0.0) | 1.9 (+1.4/–0.2) | 5.4 (+4.1/–0.5) |
| CPC | <0.1 | <0.1 | <0.1 |

**Table 2:** Particle number emission indices (EIs) (particles kg-fuel$^{-1}$) and sub-10 nm fractions.

| | 25 percentile | 50 percentile | 75 percentile |
|---|---|---|---|
| Total | | | |
| EI($N_{2.5}$) | $8.9 \times 10^{16}$ | $1.1 \times 10^{17}$ | $1.3 \times 10^{17}$ |
| EI($N_{10}$) | $3.2 \times 10^{16}$ | $4.2 \times 10^{16}$ | $5.2 \times 10^{16}$ |
| Sub-10 nm fraction | 0.53 | 0.63 | 0.70 |
| Non-volatile | | | |
| EI($N_{2.5}$) | $2.4 \times 10^{15}$ | $5.7 \times 10^{15}$ | $1.1 \times 10^{16}$ |
| EI($N_{10}$) | $1.1 \times 10^{15}$ | $1.8 \times 10^{15}$ | $4.0 \times 10^{15}$ |
| Sub-10 nm fraction | 0.44 | 0.54 | 0.72 |

Note: The particle number EI values for unheated $N_{2.5}$, unheated $N_{10}$, 350°C heated $N_{2.5}$, and 350°C heated $N_{10}$ are referred to as total
870 EI($N_{2.5}$), total EI($N_{10}$), non-volatile EI($N_{2.5}$), and non-volatile EI($N_{10}$), respectively.