# Peer review of "Characteristics of sub-10 nm particle emissions from in-use commercial aircraft observed at Narita International Airport"

_Atmospheric Chemistry and Physics, 2020_

## Referee Comment (RC1) · Anonymous Referee #1 · 9 Jul 2020

Review of "Characteristics of sub-10 nm particle emissions from in-use commercial aircraft observed at Narita International Airport" by Takegawa et al.

This paper describes measurements of aircraft engine particle emissions during takeoff operations at Narita International Airport. Concentration measurements are made with two TSI condensation particle counters (CPC) with differing lower detection size limits ( 3 nm for the Model 3776 CPC and  7-10 nm for the Model 3771 CPC), and the difference between the particle concentrations measured by these counters is intepreted as the number concentration of sub-10-nm particles. In addition, a TSI Scanning Mobility Paricle Sizer (SMPS) and Engine Exhaust Particle Sizer (EEPS) are used to measure

the size distribution of particles. A heated tube at 350 degrees Celsius is used to remove volatile particles so that the counters and SMPS can switch between measuring all particles or only the non-volatile particle fraction. The main finding of the paper is that there are significant differences between the particle concentrations measured by the 3776 counter versus the 3771 counter. Size distribution measurements for particle sizes greater than 10 nm are also presented to support the hypothesis that a significant fraction of the total and non-volatile particle number concentrations are below 10 nm; however, as the authors note, there are substantial particle diffusional losses at these sizes and the uncertainties and data corrections are significant!

Overall, the manuscript is well written and enjoyable to read. The underlying data are available in the supplementary information, which is excellent. The paper does a great job of characterizing the detection and penetration efficiences of the particle counters (although, I have a significant quibble with the use of the 3772 CPC to characterize the latter as discussed below). I previously reviewed a prior version of this manuscript for another journal, and I'm delighted to see that the authors have incorporated many of my comments/suggestions from that review into the present manuscript.

Observational reports of aircraft engine particle emissions in the literature are fairly limited given the large diversity in aircraft/engine types and airport conditions, and thus, this study is valuable in helping to overcome the current paucity of data. The content is appropriate for Atmospheric Chemistry and Physics. The paper may be publishable, but only if the following major comments are satisfactorily addressed:

1) On Lines 21-24, Lines 314-318, and elsewhere, the manuscript implies that it is somehow significant that the total particle number exceeds the number of non-volatile particles and that the regulatory emissions are somehow not accounting for these particles. This mischaracterizes the rationale behind the engine certification testing, which is designed to evaluate the emissions contributions from different engine types under relatively controlled conditions. It is well known that the volatile particle fraction is highly variable and depends on numerous variables including the fuel sulfur content and the

environmental temperature. The regulatory focus on non-volatile particles attempts to remove some of this variability; although, there are still fuel composition impacts on soot formation that need to be accounted for. In sum, the comparison that the authors are making here is not an apples-to-apples comparison and is misleading. These sentences should be removed.

2) On Lines 25-26 it is stated that the mode diameters of the size distributions were found to be smaller than 10 nm in most cases, but this does not seem to be well established from Figure 9 (there are multiple curves where there is a discernable mode around 20 nm).

3) On Lines 27-29, it is suggested that the present paper "provides new insights into the significance of sub-10 nm particles..." that are important for human health and aviation emissions inventories. I'm not sure what these puported new insights are. The present study seems to be confirming extensive past literature that has found large emissions of volatile particles (thought to be organics and sulfuric acid), but these particles may or may not have a significant impact on health. This impact would depend on their solubility – if they are soluble, then the health impact would follow dose toxicity (which would be pretty insignificant). If they are insoluble, then they could penetrate the lungs and be important. Not all ultrafine particles are created equal here. Regarding the second point about emissions inventories – how important are these particles? They are likely to be rapidly depleted via coagulation processes, and so the number-based emissions of these sub-10 nm particles are likely to be very different even at the end of the runway as compared to the surrounding area. The strength of this statement regarding the impact of the present study needs to be toned down considerably.

4) I don't think there is support for the statement made on Line 34 that aircraft emissions somehow don't participate in any wet removal processes.

5) I agree with the authors' statement on Line 70 that the technical issues associated with particle transport and losses of sub-10 nm particles need to be properly considered. How are these technical issues addressed in the present study? On Line 128-129, it is mentioned that the diffusional loss corrections within AIM are used, but these only apply to the SMPS system itself (not the 3m sampling lines or other flow splits and particle treatments). What corrections were applied to the concentration and size distribution data? Do these corrections drive the conclusions of the present paper, or are the findings the same even if the corrections are neglected?

6) Is it reasonable to assume that the particle residuals leaving the thermaldenuder are 1 nm? What about if they were 3 nm? or 5 nm? How robust are the paper's findings to this major assumption?

7) I think it's great that the detailed removal efficiency tests described on Lines 221-222 were completed, but I question the use of the 3772 CPC as the detector since it doesn't rule out the possiblity that the particles didn't completely evaporate and would still be detectable by the 3776 CPC. If possible, it would be important to redo these experiments with the 3776 since the difference between the two CPCs is being used to infer the presence of sub-10-nm non-volatile particles.

8) I don't understand what is being referred to by the statement on Line 293 about the abscence of an "artificial nucleation mode". Please clarify.

9) On Line 381 replace "soot" with "non-volatile"

10) On Line 386, strike the "s" from the word "evidences"

11) The sentence on Lines 391-393 speculating about rapid dilution prempting the growth of soot particles is unfounded and should be removed.

12) The discussion on take-off particle number concentration impacts on aircraft cruising altitudes on Line 423 does not seem relevant to the present paper.

13) Please change the y-axis scaling for the lower-left panels of Figures 7-8 to a linear scale to clearly show the agreement between the measurements of the smaller number mode. The contribution of the larger modes are already well captured by the dV/dlogDp

plot.

14) From the inset of Figure 9a it looks like there's the beginning of a hump in the gray curves that is reflected by the black line, but in Figure 10c this doesn't seem to be the case. What is the arbitrary units scale in Figure 10c and how were these different quantitative data scaled together?

---

## Referee Comment (RC2) · Anonymous Referee #2 · 5 Aug 2020

This paper reports the characteristics of sub-10nm particle emissions from field measurements at the Narita International Airport in Tokyo, Japan. Total and non-volatile particle emissions were measured using particle counting and size distribution instruments.

The paper is well written, and includes relevant details and analysis. I found some of the observations and results presented were not put into proper context with previous findings from earlier studies reported in the literature. Recent studies were cited, but their relevance to the current work was not well stated or in some cases was overstated. There are also some inconsistencies in the description of the results which requires

clarification. I have several comments that I hope will help the authors in addressing the gaps identified.

General comments:

The introduction section needs to better state the motivation for this study. The authors provide good background and context for the study, but the motivation for investigating sub-10nm particles is lacking.

The difference between the total PM and non-volatile PM is the attributable to volatile PM. The formation of volatile PM is due to a number factors including ambient conditions, fuel used, etc. The authors while presenting data for total PM and non-volatile PM haven't made any observations about the volatile PM, which is some cases dominates.

The study reports that the sub-10nm particles are non-volatile. Why is this different from earlier studies of aircraft engine emissions at airports? Is it because lower cut-off instruments were used or the mix of aircraft compared with previous studies is different, i.e. more newer engines in the fleet with different emissions characteristics or the fuel composition was different? The authors have not discussed the key finding from the current study in the context of previous observations.

References: Include the weblink or doi for each for the references included in this paper.

Specific comments:

Lns 32-34: It is not clear how aircraft emissions are unique in this aspect compared to other transportation or emission sources. Are the authors referring to aircraft emissions during cruise? Rephrase this sentence.

Lns 38-44: I think it is important to state that direct health impacts of UFP emitted from aircraft have not been currently established.

Lns 49-51: Ambient conditions also play an important role in the formation of volatile particles. Update the text accordingly.

Lns 72-73: A specific date for the measurements is stated in the introduction, however the next section (Ln 80) lists a range. Please be consistent.

Lns 86-88: The authors state that the instruments used during the measurement at NRT have previously been used for airborne measurements. More pertinent to the discussion is how the instruments used in this study varied from earlier studies of aircraft emissions at airports.

Ln 91: Specify the make and model of the NOx detector.

Ln 131-132: The location of the EEPS was not indicated in Figure 2. Was the sample provided to the EEPS from the same inlet as that for the CPCs and SMPS? This should be stated in the manuscript.

Lns 141-142: If the scanning time of the SMPS was set to 3 minutes, it is likely that the measured size distribution would be a combination of multiple plumes and not a single event. Other studies (Herndon et al., 2008; Lobo et al., 2012; Moore et al., 2017) have shown that plumes from an individual aircraft activity/movement are on the order of less than 1 minute.

Lns 187-189: What size ranges did the different sources cover?

Ln 190: "non-volatile propane soot particles supplied from the CAST" – What were the set points? Previous studies have shown a high volatile/organic content for certain the miniCAST set points, especially with pre-mixed nitrogen (e.g. Maricq, 2014; Durdina et al., 2016).

Lns 194-195: Instead of using qualitative phrases like "somewhat longer", be specific in terms of the parameters.

Lns 218-220: It's not clear why the penetration efficiency curve at room temperature

was scaled. This should be explained and it should be noted in Figure 4 b) that these data are scaled.

Lns 233-234: This is not a fair comparison. The enhancement level for CO2 for individual plumes is not an indicator of similarity. For example, how would you compare if an air parcel has higher emissions closer to the engine but heavily diluted with ambient air vs. lower emissions closer to the engine but the plume is not as diluted? Both these cases could give the same CO2 enhancement, however, the residence time in the plume, and hence the opportunity for particles to nucleate, would be different in these two cases. Do you have any data to present this in the context of residence time in the plume?

Lns 245-252: What is the main message of this plot? Is it supposed to indicate what fraction of particles heated to 350C are below 10nm? A bar chart would be more relevant to illustrate this point.

Lns 256-257: What was the lower size cut-off for the EEPS? Was it also 10nm? Is the data in Fig 7a from the SMPS or the CPC? The text indicates CPC but the figure heading has SMPS (unheated).

Lns 270-275: This enhancement has been previously reported for measurements of exhaust plumes in the near field (Lobo et al., 2012; Timko et al., 2013; Beyersdorf et al., 2014; Trueblood et al., 2018). Based on the number and volume distributions, can any inferences be made with respect to the type of plume being sampled, i.e. idle, take-off?

Ln 293: What does "artificial nucleation mode" mean?

Lns 314-316: The SAE standard system for aircraft engine emissions measurements consists of several sections (diluter, 25m sampling line, etc) that were not included in this study. I don't think it's fair to say that the difference is between real-world conditions and regulatory measurements. The difference is between measured total particle

number and non-volatile particle number, which gives a measure of the volatile particle number emissions.

Lns 316-318: It's not clear what is meant by "standard engine tests". The SAE standard system is used for the emissions certification testing of aircraft engine emissions. The engines used in these tests do not have the wear and tear associated with in-use commercial aircraft engines. Also, the data on the nvPM emissions from the certification tests is not publicly available.

Lns 324-337: Can any inferences be drawn between the previous studies and the current one, other than the emissions being in the same range? The ambient conditions, background PM, fuel, airport operations, etc during all of these studies are different. However, the particle number emissions all fall into a similar range.

Lns 337-342: Zhang et al., 2019 did not perform any measurements themselves but used the data from previous studies in their analysis. This reference should not be included in the comparison of measurement data. Also, Zhang et al. 2019 excluded certain datasets in their analysis, and thus limits the conclusions that can be drawn from their analysis. As stated previously, there are other measurements reported from previous studies that can be used to compare and quantify the differences or similarities with the current study.

Lns 353-354: The three studies referenced here all reported bi-modal distributions. When referring to the mode diameter of particle number EIs measured downstream of the engine in the near field, a distinction between the nucleation and accumulation modes should be made. For the case here, the nucleation mode should be specified.

Lns 354-360: This discussion does not follow from the previous comparisons. The authors state that the work by Kinsey et al., 2019 is an exception, but don't state how it impacted the mode diameter of particles. Aircraft engine emissions are known to vary with fuel composition and ambient conditions, but the authors do not state the relevance of these factors to their study.

Lns 362-364: The size distributions presented thus far have been shown to be bi-modal. Why was an assumption of log-normality made? Is the constraint only for the nucleation mode? Please be specific.

Lns 386-393: While this section discusses the possible mechanisms for the production of sub-10nm particles in jet engine exhaust, it does not explain the difference observed in sub-10nm soot particles reported in earlier studies. The authors should expand upon this. Are the sub-10nm particles non-volatile metals or soot or both?

Lns 405-407: See earlier comment about real-world vs. certification emissions measurements

Technical corrections:

Ln 33: "supply" is an awkward use of the word here. Suggest changing "can supply" to "emit"

Ln 49: change "significant evolution" to "significant formation and evolution"

Lns 151-152: change "might act as" to "might contribute to"

Ln 197: change "accord" to "accordance"

Ln 203: change "after" to "downstream of"

Ln 216: change "required by" to "in"

Ln 218: change "required specification" to "requirements"

Ln 233: change "individual aircraft" to "individual aircraft movements"

Ln 377: change "researches" to "research"

Ln 386: change "evidences" to "evidence"

Ln 395: change "organic matters" to "organic matter"

Figure 3: This figure does not add any value to what has already been described in the

text. It can be removed.

Figure 4 (a): In the legend, change "specification" to "manufacturer specification"

Figure 4 (b): In the legend, change "SAE ARP6320" to "SAE ARP6320 minimum specification"

Figure 5: delete "are shown" from figure caption

Figure 10 (a) and (b): change "Unheat" to "Unheated" in the legend and in figure

References:

Beyersdorf et al., Atmos. Chem. Phys. 2014, 14, 11–23

Durdina et al., Aerosol Sci. Technol. 2016, 50, 906–918

Herndon et al., Environ. Sci. Technol. 2008, 42, 1877–1883

Maricq, Aerosol Sci. Technol. 2014, 48, 620–629

Moore et al., Sci Data 2017, 4, 170198

Lobo et al., Atmos. Environ. 2012, 61, 114–123

Timko et al., Environ. Sci. Technol. 2013, 47, 3513−3520

Trueblood et al., Atmos. Chem. Phys. 2018, 18, 17029–17045

---

## Author Comment (AC1) · 30 Sep 2020

We would like to thank the referees very much for giving us valuable comments and suggestions. We have revised the manuscript to address those comments and also made other corrections to improve the clarity of the presentation. The detailed responses to the referees' comments are attached in the Supplement.

---

## Author Response (AR1)

**Response to Referee 1**

Review of "Characteristics of sub-10 nm particle emissions from in-use commercial aircraft observed at Narita International Airport" by Takegawa et al. This paper describes measurements of aircraft engine particle emissions during takeoff operations at Narita International Airport. Concentration measurements are made with two TSI condensation particle counters (CPC) with differing lower detection size limits (3 nm for the Model 3776 CPC and 7-10 nm for the Model 3771 CPC), and the difference between the particle concentrations measured by these counters is intepreted as the number concentration of sub-10-nm particles. In addition, a TSI Scanning Mobility Paricle Sizer (SMPS) and Engine Exhaust Particle Sizer (EEPS) are used to measure the size distribution of particles. A heated tube at 350 degrees Celsius is used to remove volatile particles so that the counters and SMPS can switch between measuring all particles or only the non-volatile particle fraction. The main finding of the paper is that there are significant differences between the particle concentrations measured by the 3776 counter versus the 3771 counter. Size distribution measurements for particle sizes greater than 10 nm are also presented to support the hypothesis that a significant fraction of the total and non-volatile particle number concentrations are below 10 nm; however, as the authors note, there are substantial particle diffusional losses at these sizes and the uncertainties and data corrections are significant!

Overall, the manuscript is well written and enjoyable to read. The underlying data are available in the supplementary information, which is excellent. The paper does a great job of characterizing the detection and penetration efficiences of the particle counters (although, I have a significant quibble with the use of the 3772 CPC to characterize the latter as discussed below). I previously reviewed a prior version of this manuscript for another journal, and I'm delighted to see that the authors have incorporated many of my comments/suggestions from that review into the present manuscript.

Observational reports of aircraft engine particle emissions in the literature are fairly limited given the large diversity in aircraft/engine types and airport conditions, and thus, this study is valuable in helping to overcome the current paucity of data. The content is appropriate for Atmospheric Chemistry and Physics. The paper may be publishable, but only if the following major comments are satisfactorily addressed:

We would like to thank the referee very much for giving us valuable comments and suggestions. We have revised the manuscript to address those comments and also made other corrections to improve the clarity of the presentation. The line numbers for this response letter are based on the manuscript with track changes.

1) On Lines 21-24, Lines 314-318, and elsewhere, the manuscript implies that it is somehow significant that the total particle number exceeds the number of non-volatile particles and that the regulatory emissions are somehow not accounting for these particles. This mischaracterizes the rationale behind the engine certification testing, which is designed to evaluate the emissions contributions from different engine types under relatively controlled conditions. It is well known that the volatile particle fraction is highly variable and depends on numerous variables including the fuel sulfur content and the environmental temperature. The regulatory focus on non-volatile particles attempts to remove some of this variability; although, there are still fuel composition impacts on soot formation that need to be accounted for. In sum, the comparison that the authors are making here is not an apples-to-apples comparison and is misleading. These sentences should be removed.

As the referee pointed out, the comparison with the regulatory standard was somewhat misleading. Our measurements indicate that the median values of the total and the non-volatile $EI(N_{2.5})$ were $1.1 \times 10^{17}$ and $5.7 \times 10^{15}$ kg-fuel$^{-1}$, respectively, and the difference in these values (a factor of ~20) is interpreted as the average contribution of volatile particles. We have removed the description about the comparison with the engine exhaust measurements. We have also clarified in Section 1 that the SAE-ARP6320 has been developed for the certification of jet engine emissions and may not be directly compared with ambient measurement data.

Lines 26-31, 87-88, 433-439, 638-642

2) On Lines 25-26 it is stated that the mode diameters of the size distributions were found to be smaller than 10 nm in most cases, but this does not seem to be well established from Figure 9 (there are multiple curves where there is a discernable mode around 20 nm).

As the referee pointed out, there are multiple curves where there is a discernable mode around 20 nm. However, even for those curves, the maximum value is still found at ~10 nm in most cases (please see the plot in the next page). To quantitatively show this point, we compared the $d$EI($N$)/$d$log$D$ values between each size bin for the individual take-off plumes shown in Fig. 8. We found that the $d$EI($N$)/$d$log$D$ at the size bin of 10.8 nm exhibited the highest value for ~98% of the plumes. We also found that the $d$EI($N$)/$d$log$D$ at 10.8 nm was more than two times larger than that at 14.3 nm for 79% of the plumes. These results suggest that the $d$EI($N$)/$d$log$D$ values tended to increase with decreasing particle diameters around 10–20 nm and that the mode diameters of the $d$EI($N$)/$d$log$D$ for the nucleation mode were smaller than 10 nm for the majority of the plumes observed in this study.

Lines 516-521

3) On Lines 27-29, it is suggested that the present paper "provides new insights into the significance of sub-10 nm particles..." that are important for human health and aviation emissions inventories. I'm not sure what these puported new insights are. The present study seems to be confirming extensive past literature that has found large emissions of volatile particles (thought to be organics and sulfuric acid), but these particles may or may not have a significant impact on health. This impact would depend on their solubility – if they are soluble, then the health impact would follow dose toxicity (which would be pretty insignificant). If they are insoluble, then they could penetrate the lungs and be important. Not all ultrafine particles are created equal here. Regarding the second point about emissions inventories – how important are these particles? They are likely to be rapidly depleted via coagulation processes, and so the number-based emissions of these sub-10 nm particles are likely to be very different even at the end of the runway as compared to the surrounding area. The strength of this statement regarding the impact of the present study needs to be toned down considerably.

To our understanding, the impacts of sub-10 nm particles from aviation on climate and human health are still uncertain. Righi et al. (2013, 2016) performed global model simulations and showed that the climate impacts of aviation on aerosols were sensitive to the parameterization of nucleation-mode particles (<~20 nm) in their model. In our view, the contribution of aircraft emissions to the number concentration of nucleation or Aitken mode particles is poorly understood. Sub-10 nm particles could be efficiently deposited in the olfactory mucosa, and the subsequent translocation of solid particles along the axons of the olfactory nerve might be a concern (Oberdörster et al., 2005). The health impacts of UFPs emitted from aircraft have not been well established, as pointed out by Ohlwein et al. (2019). We have added these points in Section 1.

The primary importance of aviation-produced aerosol particles in assessing the climate impacts may be limited to the upper troposphere, and our results may not be directly transferred to the particle emissions at cruising altitudes. Nevertheless, field measurements near airports would contribute to

better understanding of aircraft emissions at cruising altitudes if they are properly integrated with engine tests and/or in-flight observations. To clarify this point, we also added the importance of integrating multiple platforms for better understanding of aircraft emissions in Section 1.

Line 102-109

4) I don't think there is support for the statement made on Line 34 that aircraft emissions somehow don't participate in any wet removal processes.

We have removed the sentence (Line 47-48).

5) I agree with the authors' statement on Line 70 that the technical issues associated with particle transport and losses of sub-10 nm particles need to be properly considered. How are these technical issues addressed in the present study? On Line 128-129, it is mentioned that the diffusional loss corrections within AIM are used, but these only apply to the SMPS system itself (not the 3m sampling lines or other flow splits and particle treatments). What corrections were applied to the concentration and size distribution data? Do these corrections drive the conclusions of the present paper, or are the findings the same even if the corrections are neglected?

The overall penetration efficiencies of particles through the sampling tubes were estimated by using the theoretical formulae proposed by Gormley and Kennedy (1949). We have added the details of the calculation procedures in Section S1 of the Supplement. Furthermore, we have added descriptions about the diffusion correction by the AIM software in Section S2 of the Supplement.

Sections S1-S2 of the Supplement

The corrections for the penetration efficiencies through the sampling tubes and the detection efficiencies were not incorporated in the UCPC and CPC data presented in Sections 3.2.1–3.2.4 because the actual size distributions in the sub-10 nm size range were uncertain. Furthermore, the corrections for the penetration efficiencies were not incorporated in the SMPS and EEPS data presented in Sections 3.2.2–3.2.3 for consistency with the UCPC/CPC data. We considered the effects of particle diffusion loss as systematic uncertainties (the error bars in Figs.6 and 7). The overall size dependency (i.e., increasing particle number concentrations with decreasing diameters below 20 nm) is consistent between the SMPS and EEPS data, regardless of the corrections.

On the other hand, we considered the penetration efficiencies and the detection efficiencies for estimating the size distributions of particle number EIs in Sections 3.2.5 and 4. This allows quantitative comparison with the particle number EIs reported by previous studies.

Lines 197-213

6) Is it reasonable to assume that the particle residuals leaving the thermal denuder are 1 nm? What about if they were 3 nm? or 5 nm? How robust are the paper's findings to this major assumption?

We estimated potential artifacts due to the nucleation of gaseous compounds vaporized from particles in the evaporation tube (nucleation artifacts). We assumed an initial cluster size of 1 nm, which approximately corresponds to the critical cluster size of sulfuric acid. As the referee pointed out, other potential artifacts may originate from the condensational growth of non-volatile particles (or residual particles downstream of the evaporation tube) smaller than the detectable size range of the UCPC (diameter < 2 nm). For example, residual particles with diameters of ~2 nm may grow to ~3 nm at an ambient mass concentration of 50 μg m$^{-3}$. However, this effect is significant only in the presence of a large fraction of non-volatile particles with diameters below 2 nm.

Lines 215-216, 249-252

7) I think it's great that the detailed removal efficiency tests described on Lines 221-222 were completed, but I question the use of the 3772 CPC as the detector since it doesn't rule out the possiblity that the particles didn't completely evaporate and would still be detectable by the 3776 CPC. If possible, it would be important to redo these experiments with the 3776 since the difference between the two CPCs is being used to infer the presence of sub-10-nm non-volatile particles.

We performed additional laboratory experiments by using the UCPC. The remaining fractions for 30 nm particles were found to be ~0.3% and <0.1% for the UCPC and CPC, respectively. The remaining fractions for 50 nm particles were found to be ~5% and <0.1% for the UCPC and CPC, respectively. These results suggested that about 5% of the 50 nm $C_{40}H_{82}$ particles might not have fully vaporized but shrunk to sizes between 2.5 and 10 nm downstream of the evaporation tube.

However, under the field measurement conditions, residues of >50-nm particles were likely negligible compared with the observed sub-10 nm non-volatile particles because the particle number concentrations for a diameter range of >50 nm measured by the EEPS (unheated) were far below the observed values of the 350°C heated $N_{2.5} - N_{10}$ (Fig. 7a). In addition, residues of 30–50 nm particles in the sub-10 nm size range after the evaporation tube were likely minor compared with the observed sub-10 nm non-volatile particles considering the sharp decrease in the $dN/d\log D$ values from 30 to 50 nm as measured by the EEPS (the remaining fractions were ~0.3% and 2% for 30 and 43 nm $C_{40}H_{82}$ particles, respectively).

We appreciate the referee very much for encouraging us to perform additional laboratory experiments. We consider that the discussion has become clearer compared to the previous version.

Lines 278-285, 307-320

8) I don't understand what is being referred to by the statement on Line 293 about the abscence of an "artificial nucleation mode". Please clarify.

We meant that the size dependency of non-volatile particles (gradual increasing trends with decreasing particle diameters from 100 to 15 nm) is unlikely to be explained by the artificial growth of particles downstream of the evaporation tube because the mass concentrations of aerosols in the plumes would not have been sufficient to yield particle growth exceeding 1 nm.

Lines 406-411

9) On Line 381 replace "soot" with "non-volatile"

Corrected (Line 587).

10) On Line 386, strike the "s" from the word "evidences"

Corrected (Line 585).

11) The sentence on Lines 391-393 speculating about rapid dilution prempting the growth of soot particles is unfounded and should be removed.

We have removed the sentence (Lines 597-599).

12) The discussion on take-off particle number concentration impacts on aircraft cruising altitudes on Line 423 does not seem relevant to the present paper.

This is related to the comment 3). As the referee pointed out, the results of this study may not be directly transferred to the particle emissions at cruising altitudes. We have moved this paragraph to the end of Section 4.2 (discussion) as an implication. Nevertheless, field measurements near airports would contribute to better understanding of aircraft emissions at cruising altitudes if they are properly integrated with engine tests and/or in-flight observations.

Lines 622-630

To clarify this point, we also added the importance of integrating multiple platforms for better understanding of aircraft emissions in Section 1. Field measurements of advected (diluted) aircraft exhaust plumes near runways are not optimal for obtaining systematic emission data, whereas potential artifacts associated with long sampling lines and/or high concentrations of condensable materials can be reduced by this approach. Furthermore, a variety of exhaust plumes from different types of in-use aircraft engines can be collectively characterized by field measurements near runways. Considering that the accessibility to platforms for sampling fresh engine exhausts (engine test cells, aircraft hangers, and runways) is generally restricted, these approaches should be complementarily selected for better characterizing UFP emissions from aircraft. Consistent integration of the data is also important for constructing reliable emission inventories from the

aviation sectors for the global troposphere (cruising altitudes), where the accessibility to sampling platforms is extremely limited.

Lines 102-109

13) Please change the y-axis scaling for the lower-left panels of Figures 7-8 to a linear scale to clearly show the agreement between the measurements of the smaller number mode. The contribution of the larger modes are already well captured by the dV/dlogDp

We have changed the y-axis for the total particle $dN/d\log D$ to a linear scale, as the referee suggested. We would like to keep the log scale for the non-volatile particle $dN/d\log D$ because the absolute values differed by an order of magnitude between the EEPS and SMPS.

Figures 6 and 7 in the revised version

14) From the inset of Figure 9a it looks like there's the beginning of a hump in the gray curves that is reflected by the black line, but in Figure 10c this doesn't seem to be the case. What is the arbitrary units scale in Figure 10c and how were these different quantitative data scaled together?

We have changed the sensitivity calculations to obtain quantitative estimates of the $d\text{EI}(N)/d\log D$. The GMD and GSD values assumed in Eq. (4) were used for the $d\text{EI}(N)/d\log D$ in Eq. (5). The calculated $\text{EI}(N_{2.5})$ was compared with the median $\text{EI}(N_{2.5})$ derived from the observations to retrieve the absolute values of $d\text{EI}(N)/d\log D$ and $\text{EI}(N)$. The retrieved $d\text{EI}(N)/d\log D$ values were found to be consistent with those derived from the EEPS (Fig. 8), which supports the validity of our estimate.

We appreciate the referee very much for this comment. We consider that the discussion has become clearer as compared to that in the previous version.

Line 540-576, Figures 9 and 10

**Response to Referee 2**

This paper reports the characteristics of sub-10nm particle emissions from field measurements at the Narita International Airport in Tokyo, Japan. Total and non-volatile particle emissions were measured using particle counting and size distribution instruments.

The paper is well written, and includes relevant details and analysis. I found some of the observations and results presented were not put into proper context with previous findings from earlier studies reported in the literature. Recent studies were cited, but their relevance to the current work was not well stated or in some cases was overstated. There are also some inconsistencies in the description of the results which requires clarification. I have several comments that I hope will help the authors in addressing the gaps identified.

We would like to thank the referee very much for giving us valuable comments and suggestions. We have revised the manuscript to address those comments and also made other corrections to improve the clarity of the presentation. The line numbers for this response letter are based on the manuscript with track changes.

General comments:
The introduction section needs to better state the motivation for this study. The authors provide good background and context for the study, but the motivation for investigating sub-10nm particles is lacking.

To our understanding, the impacts of sub-10 nm particles from aviation on climate and human health are still uncertain. Righi et al. (2013, 2016) performed global model simulations and showed that the climate impacts of aviation on aerosols were sensitive to the parameterization of nucleation-mode particles ($<\sim$20 nm) in their model. In our view, the contribution of aircraft emissions to the number concentration of nucleation or Aitken mode particles is poorly understood. Sub-10 nm particles could be efficiently deposited in the olfactory mucosa, and the subsequent translocation of solid particles along the axons of the olfactory nerve might be a concern (Oberdörster et al., 2005). The health impacts of UFPs emitted from aircraft have not been well established, as pointed out by Ohlwein et al. (2019). We have added these points in Section 1. Line 49-68

The primary importance of aviation-produced aerosol particles in assessing the climate impacts may be limited to the upper troposphere, and our results may not be directly transferred to the particle emissions at cruising altitudes. Nevertheless, field measurements near airports would contribute to better understanding of aircraft emissions at cruising altitudes if they are properly integrated with engine tests and/or in-flight observations. To clarify this point, we also added the importance of integrating multiple platforms for better understanding of aircraft emissions in Section 1.

Line 102-109

The difference between the total PM and non-volatile PM is the attributable to volatile PM. The formation of volatile PM is due to a number factors including ambient conditions, fuel used, etc. The authors while presenting data for total PM and non-volatile PM haven't made any observations about the volatile PM, which is some cases dominates.

As the referee pointed out, the particle number EIs were dominated by volatile particles. This point has been clarified in Sections 3.2.4 and 5. Based on offline chemical analysis, we have previously shown that jet engine lubrication oil was the major source of aerosol particles with diameters ranging from ~10 to 30 nm in air parcels observed during the measurement period (Fushimi et al., 2019). However, detailed analysis on the formation mechanisms of volatile particles was not performed in this study. This would be an important topic in future studies

Lines 437-439, 640-642

The study reports that the sub-10nm particles are non-volatile. Why is this different from earlier studies of aircraft engine emissions at airports? Is it because lower cut-off instruments were used or the mix of aircraft compared with previous studies is different, i.e. more newer engines in the fleet with different emissions characteristics or the fuel composition was different? The authors have not discussed the key finding from the current study in the context of previous observations.

The presentation of the results was ambiguous in the previous version. The major points of this manuscript are as follows:

- The median values of the total and the non-volatile EI($N_{2.5}$), which likely cover the major size range of aircraft emissions, were found to be $1.1 \times 10^{17}$ and $5.7 \times 10^{15}$ kg-fuel$^{-1}$, respectively. The difference in these values (a factor of ~20) is interpreted as the average contribution of volatile particles.
- More than half the total and the non-volatile particle number EIs in the aircraft take-off plumes were found in the size range smaller than 10 nm.

Therefore, the sub-10 nm particles were mostly volatile. The significance of sub-10 nm size ranges for the total particles was qualitatively consistent with previous studies, but that for the non-volatile particles was unexpected.

Lines 638-651

The key point in our results is that the non-volatile particle number and volume EIs originating from soot-mode particles (>20 nm) were much smaller than those reported by the previous studies for take-off plumes under real-world operating conditions (Lobo et al., 2012, 2015b; Moore et al., 2017a). This feature might be related to the significance of sub-10 nm non-volatile particles. The data presented by Moore et al. (2017a), which were obtained in 2014, exhibited lower contributions of soot-mode particles compared to those by Lobo et al. (2012, 2015b), which were obtained in

2005 and 2004, respectively. A possible explanation for this tendency is that newer engines might emit smaller non-volatile particles as compared with older engines, as the referee suggested.

Lines 607-613

The cut-off size of instruments is an important factor for the interpretation of the data, as the referee pointed out. However, detailed discussion on this point was not made because the information on the size dependent detection efficiencies for the instruments used in the previous studies were not available.

References: Include the weblink or doi for each for the references included in this paper.

We have added the doi numbers.

Specific comments:

(1) Lns 32-34: It is not clear how aircraft emissions are unique in this aspect compared to other transportation or emission sources. Are the authors referring to aircraft emissions during cruise? Rephrase this sentence.

We have removed the sentence (Lines 47-48).

(2) Lns 38-44: I think it is important to state that direct health impacts of UFP emitted from aircraft have not been currently established.

We have added this point (Lines 67-68). Please also see the answer regarding the significance of sub-10 nm particles (General comments).

(3) Lns 49-51: Ambient conditions also play an important role in the formation of volatile particles. Update the text accordingly.

Corrected (Line 76).

(4) Lns 72-73: A specific date for the measurements is stated in the introduction, however the next section (Ln 80) lists a range. Please be consistent.

Corrected (Lines 110-111).

(5) Lns 86-88: The authors state that the instruments used during the measurement at NRT have previously been used for airborne measurements. More pertinent to the discussion is how the instruments used in this study varied from earlier studies of aircraft emissions at airports.

The UCPC and CPC used for the field measurements were exactly the same as those used for airborne measurements in our previous studies, except for the dilution/heater sections.

Lines 125-127

(6) Ln 91: Specify the make and model of the NOx detector.

Corrected (Lines 130-131).

(7) Ln 131-132: The location of the EEPS was not indicated in Figure 2. Was the sample provided to the EEPS from the same inlet as that for the CPCs and SMPS? This should be stated in the manuscript.

The EEPS was operated independently from the UCPC/CPC/SMPS inlet system and it measured the unheated particle number size distributions during the entire period. The schematic of the inlet was added in Figure 2b.

Lines 178-183, Figure 2

(8) Lns 141-142: If the scanning time of the SMPS was set to 3 minutes, it is likely that the measured size distribution would be a combination of multiple plumes and not a single event. Other studies (Herndon et al., 2008; Lobo et al., 2012; Moore et al., 2017) have shown that plumes from an individual aircraft activity/movement are on the order of less than 1 minute.

The scanning time of the SMPS is an important issue. As described in Section 3.2.2, the time periods were carefully selected for investigating the effects of rapid changes in the particle number concentration during each SMPS scan on the derived particle size distribution. The SMPS needed ~30 s to scan the major population of particle number concentrations in the aircraft plumes (<30 nm). The EEPS data were averaged over 40 s around the timing of the SMPS scan corresponding to the diameter range of 15–30 nm (i.e., from 10 s before to 30 s after the onset of each SMPS scan).

Lines 361-366

(9) Lns 187-189: What size ranges did the different sources cover?

We have added the information on the particle size (EAG: 6–15 nm, CAST: 15–100 nm; and tube furnace particle generator: 15–30 nm). Lines 262–265

(10) Ln 190: "non-volatile propane soot particles supplied from the CAST" – What were the set points? Previous studies have shown a high volatile/organic content for certain the miniCAST set points, especially with pre-mixed nitrogen (e.g. Maricq, 2014; Durdina et al., 2016).

The CAST at AIST does not provide detailed information on the mixing ratios of gases (we just set the particle diameter). As the referee pointed out, internal mixtures of organic matters might be a concern. We put a tube furnace downstream of the CAST to remove organic compounds internally or externally mixed with soot. We used a volatility tandem DMA with a heater temperature of 350°C to confirm that the thermal treatment efficiently removed organic compounds; i.e., there was no significant difference in the mobility diameter of soot particles between the two DMAs.

Lines 265-268

(11) Lns 194-195: Instead of using qualitative phrases like "somewhat longer", be specific in terms of the parameters.

The length of the sampling line for the CPC was ~90 cm. We have added this information.

Line 271

(12) Lns 218-220: It's not clear why the penetration efficiency curve at room temperature was scaled. This should be explained and it should be noted in Figure 4 b) that these data are scaled.

The theoretical formulae proposed by Gormley and Kennedy (1949) requires the temperature profile in the sample air. Because the actual temperature profile of the sample air was uncertain at 350°C, it turned out to be difficult to perform a simple theoretical prediction of the penetration efficiency at 350°C. The description in the previous version was somewhat confusing. The penetration efficiency curve at 350°C (not shown in Fig. 3) was determined by scaling the calculated curve at 20°C with the experimental values at 350°C and at room temperature (~19–21°C) for larger diameters (>30, 50, and 100 nm).

Lines 301-306, Caption of Fig. 3

(13) Lns 233-234: This is not a fair comparison. The enhancement level for CO2 for individual plumes is not an indicator of similarity. For example, how would you compare if an air parcel has higher emissions closer to the engine but heavily diluted with ambient air vs. lower emissions closer to the engine but the plume is not as diluted? Both these cases could give the same CO2 enhancement, however, the residence time in the plume, and hence the opportunity for particles to nucleate, would be different in these two cases. Do you have any data to present this in the context of residence time in the plume?

The wind directions were east or east-southeast for Fig. 4a (average wind speed: $3.9 \pm 0.7$ m s$^{-1}$) and east or east-northeast for Fig. 4b (average wind speed: $4.3 \pm 0.6$ m s$^{-1}$), indicating that relatively stable winds from the direction of the runway were dominant. The ambient temperature was ~10°C for Fig. 4a and ~8°C for Fig. 4b. Although the characteristics of the plumes shown in Figs. 4a and 4b cannot be directly compared, the similarity in the wind conditions suggests that the ages of the plumes were comparable (~50 s for the plumes in Fig. 4a and ~45 s for those in Fig. 4b).

Lines 326-339

(14) Lns 245-252: What is the main message of this plot? Is it supposed to indicate what fraction of particles heated to 350C are below 10nm? A bar chart would be more relevant to illustrate this point.

We have added the median values of the $N_{10}/N_{2.5}$ ratios for each concentration bin (Fig. 5b). The scatterplot would be still useful to show the variability in the ratios. Therefore, we would like to keep the scatterplot in addition to the median plot.

Lines 352-353, Fig. 5

(15) Lns 256-257: What was the lower size cut-off for the EEPS? Was it also 10nm? Is the data in Fig 7a from the SMPS or the CPC? The text indicates CPC but the figure heading has SMPS (unheated).

Although the particle diameter range detectable by the EEPS extended from ~6 to 520 nm, our laboratory experiments showed that the EEPS may significantly underestimate particle number concentrations below 10 nm. Reduced detection efficiencies of sub-10 nm particles can also be inferred from the EEPS data obtained by other researchers (Moore et al., 2017a). In this study we used the EEPS data for particle diameters larger than 10 nm. Therefore, the EEPS and CPC data may be directly compared. We have modified the figure heading accordingly.

Lines 183-188, Fig. 6

(16) Lns 270-275: This enhancement has been previously reported for measurements of exhaust plumes in the near field (Lobo et al., 2012; Timko et al., 2013; Beyersdorf et al., 2014; Trueblood et al., 2018). Based on the number and volume distributions, can any inferences be made with respect to the type of plume being sampled, i.e. idle, take-off?

In the enhancement event shown here, the particle volume size distributions appeared to be bimodal, but the larger mode was significantly affected by accumulation-mode particles in the background air. As described in the Supplement, the distance from the observation point to the taxiway was ~380 m and that to the gate was >800 m. We expected that aircraft emissions during idling would contribute to relatively broad, diffuse increases in aerosols and $CO_2$, which would be difficult to characterize. We also expect that aircraft emissions during take-off and landing would appear as spiked increases in aerosols and $CO_2$ at the observation point. Therefore, we assume that the observed enhancements of aerosols were mostly from take-off or landing. To avoid this ambiguity, we have extracted discrete take-off plumes and subtracted the background contributions (Fig. 8).

The references that the referee provided are helpful (Lobo et al., 2012; Timko et al., 2013; Beyersdorf et al., 2014; Trueblood et al., 2018). We have carefully checked these papers, and selected Lobo et al. (2012) as a suitable reference in this context. The sampling setup (deployed instruments and sampling location relative to the runways) and the analysis procedures (discrete plume analysis) of this study are similar to those of Lobo et al. (2012, 2015b) and Moore et al. (2017a). For the similarities and differences between those studies and our results, please see the answer to the comment (20).

(17) Ln 293: What does "artificial nucleation mode" mean?

We meant that the size dependency of non-volatile particles (gradual increasing trends with decreasing particle diameters from 100 to 15 nm) is unlikely to be explained by the artificial growth of particles downstream of the evaporation tube because the mass concentrations of aerosols in the

plumes would not have been sufficient to yield particle growth exceeding 1 nm.

Lines 406-411

(18) Lns 314-316: The SAE standard system for aircraft engine emissions measurements consists of several sections (diluter, 25m sampling line, etc) that were not included in this study. I don't think it's fair to say that the difference is between real-world conditions and regulatory measurements. The difference is between measured total particle number and non-volatile particle number, which gives a measure of the volatile particle number emissions.

As the referee pointed out, the comparison with the regulatory standard was somewhat misleading. Our measurements indicate that the median values of the total and the non-volatile $EI(N_{2.5})$ were $1.1 \times 10^{17}$ and $5.7 \times 10^{15}$ kg-fuel$^{-1}$, respectively, and the difference in these values (a factor of ~20) is interpreted as the average contribution of volatile particles. We have removed the description about the comparison with the engine exhaust measurements. We have also clarified in Section 1 that the SAE-ARP6320 has been developed for the certification of jet engine emissions and may not be directly compared with ambient measurement data.

Lines 26-31, 87-88, 433-439, 638-642

(19) Lns 316-318: It's not clear what is meant by "standard engine tests". The SAE standard system is used for the emissions certification testing of aircraft engine emissions. The engines used in these tests do not have the wear and tear associated with in-use commercial aircraft engines. Also, the data on the nvPM emissions from the certification tests is not publicly available.

We initially considered that our non-volatile particle measurements can be compared with the engine exhaust measurements by the SAE protocol. However, as mentioned in the previous answer, it is not appropriate to refer the engine certification tests here. We have removed this sentence.

Lines 433-437

(20) Lns 324-337: Can any inferences be drawn between the previous studies and the current one, other than the emissions being in the same range? The ambient conditions, background PM, fuel, airport operations, etc during all of these studies are different. However, the particle number emissions all fall into a similar range.

As described in the answer to the comment (16), the sampling setup (deployed instruments and sampling location relative to the runways) and the analysis procedures (discrete plume analysis) of this study are similar to those of Lobo et al. (2012, 2015b) and Moore et al. (2017a). Therefore, it is worthwhile discussing the similarities and differences between those studies and our results.

The fuel sulfur content is an important parameter for the comparison with other studies. We do not have information on the sulfur content of the fuel that was actually used at NRT. Instead, we

analyzed fuel samples (Jet A-1) provided by a jet fuel company in Tokyo (Ishinokoyu, Co. Ltd.) (Saitoh et al., 2019b). The sulfur content of the fuel samples ranged from 30.4 to 440 parts per million by weight (ppmw). We assume that these values are representative of the sulfur content of jet fuels commercially available in Tokyo during the observation period.

Lobo et al. (2012) reported particle number and mass EIs measured 100–350 m downwind of the runways at Oakland International Airport in August 2005. The total particle number EIs for various types of engines under take-off conditions ranged from $4 \times 10^{15}$ to $2 \times 10^{17}$ kg-fuel$^{-1}$, which inclusively covered the 25–75 percentile range of the total EI($N_{2.5}$) from our measurements. The fuel sulfur content was estimated to be 240–395 ppmw.

Lobo et al. (2015b) reported the particle number and mass EIs measured near the jet engine exits and 100–350 m downwind of the runways at Hartsfield-Jackson Atlanta International Airport in September 2004. The total particle number EIs for various types of engines under take-off conditions ranged from $7 \times 10^{15}$ to $9 \times 10^{17}$ kg-fuel$^{-1}$, which again inclusively covered the 25-75 percentile range of total EI($N_{2.5}$) from our measurements. The information on the fuel sulfur content was not provided.

Moore et al. (2017a) reported the particle number and volume EIs for take-off plumes based on field observations conducted at Los Angeles International Airport (LAX) in May 2014. We calculated the median values (and the 25–75 percentile range) of the total and the non-volatile particle number EIs provided by Moore et al. (2017a) to be 4.6 (3.1–5.8) $\times 10^{16}$ and 2.1 (1.1–3.6) $\times 10^{15}$ kg-fuel$^{-1}$, respectively. The median and 25–75 percentile range of the total and the non-volatile EI($N_{10}$) from our measurements showed good agreement with those from Moore et al. (2017a). The sulfur content of the jet fuel samples collected at LAX ranged from 620 to 1,780 ppmw (average: 1,180 ppm).

Timko (2010) showed that the total particle number EIs in moderately diluted plumes exhibited a relatively weak dependence on the fuel sulfur content (<1,500 ppmw) under high thrust conditions for various types of engines. Provided that the fuel sulfur content was likely below ~1,500 ppmw for the previous studies (Lobo et al., 2012, 2015b; Moore et al., 2017a) and this study, it might not be the major factor affecting the variability in the emissions of volatile particles among these studies.

We consider that the referee's question is very important. The total and non-volatile particle number EIs derived from the UCPC and CPC fell in the same range as those from the previous studies for take-off plumes under real-world operating conditions (Lobo et al., 2012, 2015b; Moore et al., 2017a). However, the characteristics of the size distributions appeared to be significantly different. Specifically, the non-volatile particle number and volume EIs originating from soot-mode particles

(>20 nm) were much smaller than those reported by the previous studies. This feature might be related to the significance of sub-10 nm non-volatile particles. The data presented by Moore et al. (2017a), which were obtained in 2014, exhibited lower contributions of soot-mode particles compared to those by Lobo et al. (2012, 2015b), which were obtained in 2005 and 2004, respectively. A possible explanation for this tendency is that newer engines might emit smaller non-volatile particles as compared with older engines.

We have added the above points in Sections 3.2.4 and 4.2. We appreciate the referee very much for this comment. We consider that the discussion has become clearer.

Lines 459-492, 580-613

(21) Lns 337-342: Zhang et al., 2019 did not perform any measurements themselves but used the data from previous studies in their analysis. This reference should not be included in the comparison of measurement data. Also, Zhang et al. 2019 excluded certain datasets in their analysis, and thus limits the conclusions that can be drawn from their analysis. As stated previously, there are other measurements reported from previous studies that can be used to compare and quantify the differences or similarities with the current study.

We have removed Zhang et al. (2019) from the comparison (Lines 492-496).

(22) Lns 353-354: The three studies referenced here all reported bi-modal distributions. When referring to the mode diameter of particle number EIs measured downstream of the engine in the near field, a distinction between the nucleation and accumulation modes should be made. For the case here, the nucleation mode should be specified.

We have specified the nucleation mode when we discuss the mode diameter (Lines 514-523).

(23) Lns 354-360: This discussion does not follow from the previous comparisons. The authors state that the work by Kinsey et al., 2019 is an exception, but don't state how it impacted the mode diameter of particles. Aircraft engine emissions are known to vary with fuel composition and ambient conditions, but the authors do not state the relevance of these factors to their study.

As the referee pointed out, the discussion does not follow from the previous comparisons. The results from Kinsey et al. (2019) might not be directly compared with our results. We have removed the comparison.

Lines 531-537

(24) Lns 362-364: The size distributions presented thus far have been shown to be bimodal. Why was an assumption of log-normality made? Is the constraint only for the nucleation mode? Please be specific.

We assumed monomodal size distributions for $dN/d\log D$ and $dEI(N)/d\log D$ considering the shapes of the observed size distributions for total and non-volatile particles (Figs. 6 and 7). It is difficult to retrieve bimodal size distributions from the observations because the particle number concentrations were dominated by the smaller mode (we would not use "nucleation mode" for non-volatile particles because it might lead to confusion).

Lines 552-554

(25) Lns 386-393: While this section discusses the possible mechanisms for the production of sub-10nm particles in jet engine exhaust, it does not explain the difference observed in sub-10nm soot particles reported in earlier studies. The authors should expand upon this. Are the sub-10nm particles non-volatile metals or soot or both?

We have removed the discussion on the soot formation mechanisms because it was too speculative. Currently we do not have a direct evidence that the observed sub-10 nm non-volatile particles are composed mainly of soot. Our estimate implies the presence of very small non-volatile particles with diameters down to a few nanometers. This is not consistent with the size of the primary soot particles from jet engines estimated by using transmission electron microscopy (TEM) or laser-induced incandescence (LII) methods (e.g., Liati et al., 2014; Boies et al., 2015; Saffaripour et al., 2017, 2020).

An alternative possibility for the significance of sub-10 nm non-volatile particles includes the potential contributions of less volatile organic matter (as compared with $C_{40}H_{82}$) or metal compounds. Fushimi et al. (2019) showed that the mass contribution of the sum of trace elements (other than carbonaceous and sulfur compounds) was comparable to that of elemental carbon (soot) for UFP samples (~10–30 nm) collected at NRT. Saitoh et al. (2019a) reported that metal elements including Ca, Fe, Si, Mg, K, Zn, Pb, and Ni were the major compositions of these trace elements.

Lines 584-606

(26) Lns 405-407: See earlier comment about real-world vs. certification emissions measurements

Please see the answer to the comment (19).

(27) Technical corrections:

Ln 33: "supply" is an awkward use of the word here. Suggest changing "can supply" to "emit"

Ln 49: change "significant evolution" to "significant formation and evolution"

Lns 151-152: change "might act as" to "might contribute to"

Ln 197: change "accord" to "accordance"

Ln 203: change "after" to "downstream of"

Ln 216: change "required by" to "in"

Ln 218: change "required specification" to "requirements"

Ln 233: change "individual aircraft" to "individual aircraft movements"

Ln 377: change "researches" to "research"

Ln 386: change "evidences" to "evidence"

Ln 395: change "organic matters" to "organic matter"

Figure 3: This figure does not add any value to what has already been described in the text. It can be removed.

Figure 4 (a): In the legend, change "specification" to "manufacturer specification"

Figure 4 (b): In the legend, change "SAE ARP6320" to "SAE ARP6320 minimum specification"

Figure 5: delete "are shown" from figure caption

Figure 10 (a) and (b): change "Unheat" to "Unheated" in the legend and in figure

We have corrected the above points. We appreciate the referee for detailed proofreading. We have removed Fig. 3 following the referee's comment.

We thank the referee for providing the information. We have added some of the references in the revised version.

[revised manuscript text omitted]
 (1949). The calculations assumed a laminar flow at a pressure of 101 kPa and a temperature of 293 K. The inlet system was divided into the subsections listed in Table S1, and the overall penetration efficiency was derived as a product of the penetration efficiencies through the subsections. The subsection "UCPC internal" in Table S1 represents the effective tube length that can reproduce the penetration efficiency through a UCPC 3776 (Wimmer et al., 2013). The detection efficiencies of the UCPC and CPC, which are denoted as $\varepsilon_{\text{UCPC}}$ and $\varepsilon_{\text{CPC}}$, were set as follows based on our previous studies (Takegawa and Sakurai, 2011; Takegawa et al., 2017):

$$\varepsilon_{\text{UCPC}} = \left(1 + \exp\left(\frac{2.50 - D}{0.111}\right)\right)^{-1}$$

$$\varepsilon_{\text{CPC}} = 1 - \exp\left(\frac{6.99 - D}{4.78}\right)$$

where $D$ is the particle diameter. Note that the detection efficiency for the UCPC was empirically determined so as to satisfy $\varepsilon_{\text{UCPC}} = \sim 0,\ 0.5,\ \sim 1$ at diameters of $<2,\ 2.5,\ >3$ nm, respectively. This assumption does not significantly affect the major conclusions because the contributions of particles at 2–3 nm were minor in the theoretical calculations (Figs. 9 and 10). As shown in Fig. 3, the detection efficiencies of the UCPC and CPC incorporating the penetration efficiencies of the subsections agreed well with the experimental data.

Fig. S1 shows the penetration efficiency for the UCPC/CPC sampling line under the unheated mode (UCPC/CPC main × UCPC/CPC total sample × Unheated), that for the SMPS sampling line under the unheated mode (UCPC/CPC main × UCPC/CPC total sample × Unheated × SMSP sample), and that for the EEPS sampling line (EEPS main × EEPS sample). The penetration efficiency through the evaporation tube and the detection efficiencies of the UCPC and CPC were evaluated separately from the above estimates (see Section 3.1).

**Table S1:** Parameters for calculating the penetration efficiencies of particles for the UCPC, CPC, SMPS, and EEPS.

| Subsection | Flow rate (L min⁻¹) | Length (cm) | Penetration efficiency at 10 nm |
|---|---|---|---|
| UCPC/CPC main (rooftop – branch) | 20 | 280 | 0.97 |
| UCPC/CPC total sample (branch – mixing junction) | 0.7 | 30 | 0.94 |
| Unheated (mixing junction – splitter) | 2.7 | 67 | 0.96 |
| Heated (mixing junction – heater – splitter) | 2.7 | 152 | 0.92 |
| UCPC sample (splitter – UCPC) | 1.4 | 50 | 0.94 |
| CPC sample (splitter – CPC) | 1.0 | 90 | 0.90 |
| SMPS sample (splitter – SMPS) | 0.3 | 100 | 0.77 |
| UCPC internal | 0.3 | 19 | 0.92 |
| EEPS main (rooftop – manifold branch) | 20 | 276 | 0.97 |
| EEPS sample (manifold branch – EEPS) | 10 | 100 | 0.97 |

Note: See Fig. 2 for the schematics of the subsections. The flow rate and length of each section are approximate values.

[Figure]

**Figure S1:** Penetration efficiencies of particles through the sampling lines for the UCPC/CPC (red), SMPS (blue), and EEPS (black) at room temperature (293 K) estimated by using the theoretical formulae proposed by Gormley and Kennedy (1949). The detectable size ranges for the UCPC, SMPS, and EEPS are indicated by solid lines. The calculated curve for the UCPC/CPC includes the penetration efficiency through the UCPC/CPC main, UCPC/CPC total sample, and Unheated subsections. The calculated curve for the SMPS includes the penetration efficiencies through the UCPC/CPC main, UCPC/CPC total sample, Unheated, and SMSP sample subsections. The calculated curve for the EEPS includes the penetration efficiencies through the EEPS main and EEPS sample subsections.

**S2 Diffusion correction for the SMPS**

40      The Aerosol Instrument Manager (AIM) software provides correction tools for these factors. We used AIM version 9.0 for the present analysis, with the diffusion loss correction enabled. Fig. S2 shows a comparison of the SMPS size distribution at 14:10 on February 15, 2018 with and without the AIM diffusion correction. The degree of correction was significant at smaller diameters. Nevertheless, the overall size dependency (i.e., increasing particle number concentrations with decreasing particle diameters below 20 nm) is consistent between the SMPS and EEPS data, regardless of the AIM diffusion correction.

[Figure]

**Figure S2:** Particle number size distributions measured simultaneously by the EEPS and unheated SMPS (with and without the AIM diffusion correction) for the selected time period indicated in Fig. 6.

**S3 Plume analysis**

In the observed air parcels, aircraft emissions from take-off, landing, and idling phases may have been mixed in the atmosphere, and the characterization of particle emissions should be performed carefully. The distance from the observation point to the taxiway was ~380 m and that to the gate was >800 m. We expected that aircraft emissions during idling would contribute to relatively broad, diffuse increases in aerosols and $CO_2$ and that those during take-off and landing would appear as spiked increases in aerosols and $CO_2$ at the observation point.

To extract discrete plumes originating from individual aircraft during take-off or landing, we defined background levels for $N_{2.5}$, $N_{10}$, and $CO_2$, and calculated enhancements above the background levels ($\Delta N_{2.5}$, $\Delta N_{10}$, and $\Delta CO_2$). The background estimate is more critical for $CO_2$. For air parcels originating from the runway (wind directions from north to east-southeast, wind speeds of >1 m s$^{-1}$), the sets of air parcels that were selected by the following procedures were defined as "plumes":

(a) The background air was defined as satisfying the following conditions: $| \, dCO_2/dt \, | < 0.1$ ppmv s$^{-1}$, $| \, d^2CO_2/dt^2 \, | < 0.1$ ppmv s$^{-2}$, $| \, dN_{10}/dt \, | < 500$ cm$^{-3}$ s$^{-1}$, and $N_x < N_{th}$, ($x = 2.5$ or 10) where $d/dt$ represents the time differential. The second and fourth conditions were set to exclude plume peaks. The threshold value, $N_{th}$, depended on the meteorological conditions and was set to an appropriate values for each day.

(b) The above background values were interpolated to determine the baselines for $N_{2.5}$, $N_{10}$, and $CO_2$. The baseline was subtracted to obtain $\Delta N_{2.5}$, $\Delta N_{10}$, and $\Delta CO_2$.

(c) If the peak $\Delta CO_2$ exceeded 15 ppmv, the $\Delta CO_2$ values decreased to below 10% of the peak value within 60 s before or after the peak, and the duration of the enhancement was longer than 30 s, the set of air parcels was selected as a "plume".

The above threshold values were determined by considering the observed shapes of the $CO_2$ and aerosol spikes. Step (a) was used to identify "stable" baseline data points, and the conditions were set as redundant. The criterion of 10% in step (c) eliminated overlaps of multiple plumes. This automated procedure may have discarded some possible plume events, depending on the meteorological condition. Nevertheless, we chose these criteria to avoid subjective biases.

Next, the $\Delta N_{2.5}/\Delta CO_2$, $\Delta N_{10}/\Delta CO_2$, and $\Delta N_{10}/\Delta N_{2.5}$ ratios for the identified plumes were calculated. Only data with $N_{10}$ smaller than $5 \times 10^5$ cm$^{-3}$ (~$1 \times 10^5$ cm$^{-3}$ downstream of the dilution section) were used for the analysis because the uncertainty due to particle coincidence increases at higher concentrations. Data obtained on February 15, 16, 20, 21, and 22 were used for the plume analysis. The $\Delta N_{2.5}/\Delta CO_2$, $\Delta N_{10}/\Delta CO_2$, and $\Delta N_{10}/\Delta N_{2.5}$ ratios were calculated by using an area-integration method, similar to that used by Moore et al. (2017a). We also calculated these ratios as linear regression slopes after the data points were averaged over 3 s. The data average was used to account for differences in the response times of the instruments. Although these two methods generally showed reasonable agreement, there were significant discrepancies in some cases, especially at low $r^2$ values by the regression method. The reason for the discrepancy at low $r^2$ values was that the temporal variations in $N_{2.5}$ and $N_{10}$ did not track well with that of $CO_2$. A possible explanation for this feature

is that particle emissions might vary significantly during take-off (e.g., a burst of soot particles in the initial stages), as pointed out by Moore et al. (2017a).

The arrival time of a plume was estimated by considering the wind directions and speeds, assuming that the time for the plume to traverse from the centerline of the runway to the observation point was controlled by the wind vector component perpendicular to the runway. The duration of a plume was estimated from the time difference between the two 10%-crossing points defined in step (c) (when the $\Delta CO_2$ values decreased to below 10% of the peak value within 60 s before or after the peak). The estimated arrival time of plumes was ~30–120 s, which corresponds to the transport distance of ~180–370 m.

The flight-schedule table provided by NRT, which specified the take-off or landing times of specific aircraft with a time resolution of 1 min, was used to investigate the statistics of aircraft take-offs and landings. During the time periods of the plume analyses, 80–90% of the aircraft that passed along the runway were in the take-off phase. The flight- schedule table, estimated arrival times, and our video-camera record (only during the daytime) were used to attribute the observed plumes to take-off or landing phases. Fig. S3 shows an example of the correspondence between the plume events and the flight information. Aerosol particle number concentrations for diameters larger than 7 nm ($N_7$) as measured by the undiluted and unheated CPC 3022 are shown for comparison. In Fig. S3a, we can see a reasonably good agreement between $N_7$ and $N_{10}$, as expected. In Fig. S3b, the depletion of aerosol particle number concentrations upon heating is evident.

Although the observed plumes could, in most cases, be attributed to the take-off or landing of specific aircraft, there were some cases in which the one-to-one correspondence was somewhat ambiguous (shown as "unidentified" in Fig S3a). We attributed 132 plumes to take-offs for the unheated mode and 63 plumes to the 350°C- heated mode. Potential uncertainties in the attribution (i.e., a landing plume incorrectly assigned to a take-off plume) were 10–20% at most, considering that 80–90% of the aircraft that passed along the runway were in the take-off phase. Table S2 shows the statistical summary of the particle number EIs classified by major aircraft models identified in this study. We did not observe a significant difference in the particle number EIs among these models, although there might be uncertainties in the attribution, as mentioned above.

Fig. S4 shows histograms of the estimated arrival time and duration of the plumes. Although our sampling conditions differed from those given by Moore et al. (2017a), the estimated arrival and duration times were comparable to their values. We did not find a systematic dependence of the $\Delta N_{10}/\Delta CO_2$ and $\Delta N_{2.5}/\Delta CO_2$ ratios on the arrival time of the plumes, as indicated in Fig. S5.

[Figure]

120    **Figure S3:** Examples of discrete plumes. Data for (a) unheated and (b) 350°C heated mode observed on February 21, 2018 . Slight differences in the peak timing for very sharp spikes may be affected by the instrument response times. The blue open circles represent the estimated "background" concentrations for $CO_2$, and the blue dashed lines represent the interpolated background levels.

[Figure]

125

Figure S4: Histograms of the estimated arrival time and duration of plumes. The lower and upper limits of the duration time (30 s and 120 s, respectively) were determined by the definition of plumes.

[Figure]

130    Figure S5: Dependence of $\Delta N_{2.5}/\Delta CO_2$ and $\Delta N_{10}/\Delta CO_2$ ratios on the arrival time of plumes.

**Table S12:** Medians of particle number EIs for take-off plumes classified by major aircraft models identified in this study ( number of samples $\geq$ 5). The unit of particle number EIs is $10^{15}$ kg-fuel$^{-1}$.

| Aircraft model | Total | | | | Non-volatile | | | |
|---|---|---|---|---|---|---|---|---|
| | Number of samples | EI($N_{2.5}$) | EI($N_{10}$) | Sub-10 nm fraction | Number of samples | EI($N_{2.5}$) | EI($N_{10}$) | Sub-10 nm fraction |
| A320 | 12 | 80 | 44 | 0.44 | N/A | N/A | N/A | N/A |
| A321 | N/A | N/A | N/A | N/A | 6 | 3.6 | 2.3 | 0.44 |
| A333 | 21 | 94 | 35 | 0.65 | 9 | 2.4 | 1.2 | 0.49 |
| B738 | 9 | 117 | 45 | 0.56 | N/A | N/A | N/A | N/A |
| B748 | 5 | 114 | 41 | 0.66 | N/A | N/A | N/A | N/A |
| B763 | 14 | 129 | 50 | 0.64 | 11 | 9.4 | 1.8 | 0.66 |
| B772 | 7 | 91 | 26 | 0.71 | N/A | N/A | N/A | N/A |
| B77W | 15 | 96 | 34 | 0.65 | 9 | 2.7 | 1.3 | 0.49 |
| B788 | 10 | 139 | 71 | 0.54 | N/A | N/A | N/A | N/A |
| B789 | 12 | 125 | 64 | 0.54 | N/A | N/A | N/A | N/A |

N/A: Not available.